# iPAR: A framework for modelling and inferring information about disease spread when the populations at risk are unknown

Stephen Catterall[1]*, Thibaud Porphyre[2], Glenn Marion[1]

**1** Biomathematics and Statistics Scotland, Edinburgh, United Kingdom, **2** Laboratoire de Biométrie et Biologie Évolutive, UMR 5558, Universite Claude Bernard Lyon 1, CNRS, VetAgro Sup, Marcy l'Étoile, France

* stephen@bioss.ac.uk

## Abstract

We introduce the inference for populations at risk (iPAR) framework which enables modelling and estimation of spatial disease dynamics in scenarios where the population at risk is unknown or poorly mapped. This framework addresses a gap in spatial infectious disease modelling approaches, with current methods typically requiring data on the spatial distribution of the population at risk. The principles for iPAR are demonstrated in the context of a susceptible-infected disease dynamics model coupled with Bayesian inference implemented via data-augmentation Markov chain Monte Carlo (MCMC). This implementation of iPAR is tested for a range of scenarios using simulated outbreak data. Results indicate that the method can effectively estimate key properties of disease spread from spatio-temporal case reports and make useful predictions of future spread. The method is then applied to a case study exploring the 2014–2019 Estonian outbreak of African Swine Fever (ASF) in wild boar. Estimates of epidemiological parameters reveal evidence for long distance transmission, as well as disease control via reduction of the wild boar population in Estonia.

## Author summary

Models for the spatial spread of infectious disease are essential tools to help us understand the way that disease spreads across a landscape during a disease outbreak. In addition to knowledge of disease case reports, most such models also assume knowledge of the spatial distribution of the disease hosts. If the distribution of the disease hosts is unknown then it becomes harder to interpret any apparent patterns of disease spread. Furthermore, most existing modelling techniques can no longer be applied. This paper introduces a novel modelling framework that can be used to analyse outbreak data even when we don't know

**Data availability statement:** The authors do not own the disease outbreak dataset used in this paper. However, the data are freely available under direct request to the World Animal Health Information System (WAHIS, https://wahis.woah.org, woah@woah.org). Computer code for performing the analyses described in this manuscript may be found at https://github.com/spcatterall/iPAR.

**Funding:** SC, GM and TP were funded by the Scottish Government Rural and Environment Science and Analytical Services Division (RESAS), as part of the Scottish Government's Centre of Expertise on Animal Disease Outbreaks (EPIC). TP gratefully acknowledges funding from the French National Research Agency and Boehringer Ingelheim Animal Health France for support through the IDEXLYON project (ANR-16-IDEX-0005) and the Industrial Chair in Veterinary Public Health, as part of the VPH Hub in Lyon. SC and GM were also supported by the Strategic Research Programme of the Scottish Government's Rural and Environment Science and Analytical Services Division (RESAS). The funders had no role in study design, data collection and analysis, decision to publish, or preparation of the manuscript.

**Competing interests:** The authors have declared that no competing interests exist.

the spatial distribution of the disease hosts. To achieve this, we exploit spatial data for the region of the outbreak, for example land use data, that can help to better inform us of the distribution of the population at risk, when combined with available disease case reports. As a case study, we explore the use of the new framework for modelling data from the 2014–2019 outbreak of African Swine Fever in wild boar in Estonia.

## Introduction

Outbreaks of infectious disease can be very costly and can have serious impacts on both natural systems and human economic activities [1,2]. For example, African swine fever (ASF), a disease that can infect both domestic pigs and wild boar, has spread since its introduction in Georgia in 2007 through many countries in Europe and Asia, leading to millions of culled pigs and consequent serious impacts on the global pig industry [3]. More broadly, including for diseases of humans and plants, epidemic models can be usefully employed to assess risks arising from both current and future disease outbreaks [4–6]. Specifically, they can be fitted to reported outbreak data to infer important characteristics of disease spread [7]. These characteristics can then be used to inform potential disease control measures [8,9].

The spatial distribution of the population at risk is very important because it strongly influences the contact structure of the population, which in turn is of key importance in the disease transmission process [10]. Further, spatially explicit epidemic models can offer insights and information critical to the control of real-world outbreaks, e.g., identifying high risk areas and control zones [11,12]. Typically, such models assume knowledge of the spatial distribution of the population at risk, e.g., foot and mouth disease (FMD) transmission models [13], Classical swine fever (CSF) transmission models [14]. However, a frequently encountered and challenging scenario is when we do not have complete knowledge of the population at risk of infection [5] and, in particular, we may not know the spatial distribution of this population. This is especially true for wildlife populations, which may not be well mapped, but can also frequently be the case for livestock. Records of livestock holdings may either be non-existent (especially for small 'backyard' holdings), out of date, incomplete or may not be publicly available for privacy reasons (e.g., farm locations in the USA [15]). In such cases there is a need to make use of case-only data. Here, we therefore focus on the scenario where cases are reported with possibly imprecise spatial and temporal coordinates.

A number of techniques have been proposed to model disease transmission when there is incomplete knowledge of the population at risk. One class of approaches involves modelling the disease case generating process, with the susceptible population not explicitly specified. Transmission trees are one example of this approach, e.g., [16]. Another example is the use of techniques based on contact distribution models [17], for example [18,19]. Such approaches benefit from the fact that, typically, more is known about disease cases than is known about general members of the susceptible

population. On the other hand, this approach may be problematic in scenarios where the set of disease case reports is incomplete, or where depletion of susceptible individuals is an important factor in determining the progression of the epidemic. There is also a risk of bias in scenarios where individuals are highly mobile, such as susceptible wildlife populations. In such scenarios the location of a case (often where an animal is found dead) may be a poor proxy for the spatial history of the individual [20]. The above referenced approaches do not allow for spatial variation in susceptible population density, with the exception of [19] in which the susceptible population density is both allowed for and known.

A second class of approaches, which has been applied to livestock epidemic data, involves making assumptions/predictions about unknown farm locations to permit spatially explicit transmission modelling. For example, assuming that farms are located uniformly at random across the landscape can sometimes be sufficient to identify optimal control measures [21], though this can only be applied to certain controls (ring culling) and requires extensive outbreak data. However, using land use data and other landscape covariates to predict farm locations can improve the accuracy of livestock epidemic models [15,22]. Such approaches seem useful in scenarios where we know some aspects of the susceptible population, e.g., for livestock the total number of farms in the region of interest, but it is unclear how they might be applied to less well mapped populations such as wildlife, 'backyard' holdings or in jurisdictions where data is more limited.

We consider how to model disease spread when the susceptible population density is unknown, but we have spatial covariates that may explain population density and other aspects of the population of relevance to the transmission process. Intuitively, case reports with approximate spatial locations are informative of the distribution of the population at risk, and coupled with observation times also provide insight into rates of spread in space and time. To motivate our proposed approach, consider again how knowledge of the population at risk is used to construct spatially explicit epidemic models, e.g., those used to model the spread of FMD [23]. The rate of transmission between an infectious location and a susceptible location is typically assumed to be an overall rate multiplied by (a) a kernel function depending on the distance between the locations, (b) the infectivity of the infectious location and (c) the susceptibility of the susceptible location. Thus, from a modelling perspective, in the most general setting, epidemiologically relevant knowledge of the population at risk is captured in two 'surfaces' covering the landscape: an infectivity surface and a susceptibility surface. In reality, although these surfaces will be influenced by the spatial distribution of the population at risk, they are also influenced by many other factors, e.g., variations in biosecurity, variations in host susceptibility. As a consequence, the two surfaces are not regarded as population density estimates for the host, but as we will see below in some cases, they can provide insights into spatial variation of the underlying population at risk. In this paper, assuming limited or no knowledge of the population at risk, we develop an approach to estimate these two surfaces along with key characteristics of spatial disease transmission, given reported disease cases, but avoiding making any explicit assumptions about the spatial distribution of the population at risk. We refer to this as the inferring - information about - populations at risk (iPAR) framework.

To constrain the set of possible surfaces, we assume that each surface is a function of multiple landscape covariates, with parameters estimated from the disease case data. We also aggregate spatially to a set of patches covering the landscape, with the patches being the infectious units of the model. This aids computational tractability but also enables use of uncertain spatial locations. Patch-based models have been used successfully to model the spread of infectious disease, the spread of invasive non-native plant species and other spatio-temporal ecological processes [24–26]. Although spatial aggregation can result in a loss of spatial resolution, the degree of aggregation (patch size) can be controlled to minimise any potential loss of information. The iPAR framework is also highly flexible and can make use of a wide array of spatial covariate data. Land use and climate data might be expected to explain at least some of the variation in patch susceptibility/infectivity. Other covariates could be useful in specific scenarios, e.g., human population density, hunting bag data and previously published estimates of the susceptible population density. The precise choice of covariates will depend on the specific application at hand and the availability of spatially explicit covariates. In the case of a human disease, relevant covariates could include local human population density and key indices, e.g., measuring social deprivation and air pollution.

In this paper we describe an initial implementation of the iPAR framework outlined above. This is described in detail in Methods along with the MCMC implementation of Bayesian inference used for parameterisation from disease case reports for the region of interest. This inference framework is extensively tested using simulated case data in Results. We illustrate the potential of the iPAR framework through application to case reports arising from the ASF outbreaks in Estonia that occurred between 2014 and 2019 in Case study. Finally, we discuss potential application to other systems and possible future development of the iPAR framework in Discussion.

## Methods

Here we define an underlying spatial disease transmission model and associated inference tools that together provide a flexible approach to deal with the problem of missing information on populations at risk in spatial epidemiology. This defines an initial implementation but note that many extensions are possible within the iPAR framework.

### The generic model

We model transmission of disease in continuous time across a region $R$. In principle, any disease could be modelled, whether of animals, plants or humans. However, the case study presented focusses on animal disease. We partition the region $R$ into a collection of $I$ patches, with each patch indexed by an integer $i$ ($1 \leq i \leq I$) and associated with a given sub-region of $R$. Within the region $R$, the spatial distribution of the population at risk is uncertain. However, we assume that we have covariates available that can be used to model the variation in patch susceptibility/infectivity across the region. In practice, we choose covariates that are likely to explain at least some of the variation in host population risk profile across the region. The patches are the units of infection within the model framework. For each patch $i$, there is a set of associated covariate values $\mathcal{C}_i$. In principle, the covariates may vary both temporally and spatially. However, for notational simplicity, we do not show explicit time dependence in the presentation that follows, and in all the applications of the methodology that we consider in this manuscript the covariates are constant in time.

The chosen size of the patches depends on a number of factors, including the spatial resolution of the covariates and the disease case data. If the covariates have a different spatial resolution to the case data then one or both data sets may be aggregated or de-aggregated to the chosen resolution. Patch size should be small enough to capture the apparent pattern of disease spread across the region.

For simplicity we consider modelling transmission according to a spatial SI model. That is, each patch is either susceptible to infection or has become infected/infectious. However, as shown in Case study, even where individual-host dynamics are not limited to susceptible and infectious compartments, patch-level SI dynamics can still provide a good approximation to the spatial transmission process. In addition, we note in general that more complex dynamics and a greater set of possible states could in principle be accounted for at the patch-scale if suitable data were available. However, such complexities are outside the scope of the current implementation. The SI assumption is most appropriate for the early stages of a disease outbreak, but can continue to be reasonable if the disease is persistent at the patch level on the timescales of the outbreak and this criterion should be considered when selecting an appropriate patch size for the analysis – for example, see Case study. The transmission of disease is modelled over some time interval $[0, T]$ via a continuous time discrete state space Markov process. The initial conditions comprise a list of the patches that are infected at time $t = 0$, with the remainder assumed susceptible.

Given the initial conditions, the transmission model is fully specified once we define the force of infection on a susceptible patch at any point in time. The force of infection

$$f_i = \varepsilon s_i + \rho \sum_j s_i t_j K(d_{ij})$$

(1)

on a susceptible patch $i$ is comprised of two terms, the first term representing primary infection due to external transmission from outside the region under study, the second term representing transmission from infected patches within the region $R$

. Each patch $i$ has susceptibility $s_i$ and, when infected, infectivity $t_i$ which can be functions of spatially varying covariates – meaning that they vary between patches. The parameter $\varepsilon$ controls the overall rate of background infection, while $\rho$ controls the overall rate of secondary transmission. The summation is over all infectious patches $j$, $K$ is a so-called transmission kernel and $d_{ij}$ is the distance between the centroids of patches $i$ and $j$. Typically, we use a truncated power law kernel $K(d) = d^{-2\lambda}I(d < d_{max})/m(\lambda)$. Here, $I$ is an indicator function (equal to 1 if its argument is a true statement, 0 otherwise), $m(\lambda)$ is a normalisation constant defined in S1 Appendix (see also [27]), $d_{max}$ is a fixed maximum transmission distance, and $\lambda$ is a parameter estimated from the outbreak data. If the patches are all squares of the same size then distance, as an input to the kernel function, is measured in units such that one unit is equal to the width of a patch, meaning that, e.g., $d_{ij} = 1$ precisely when patches $i$ and $j$ are nearest neighbours. The above expressions together represent what we call the *constant-in-time* model, although as noted above this can accommodate temporal variation in the covariates.

We also consider a variant of this model, which we term the *varying-in-time* model, in which the total force of infection is multiplied by a function of time $h(t)$. This function is intended to model the effects of disease control. For identifiability we assume that $h(0) = 1$, with $h$ then typically decreasing following the application of control measures such as removal of hosts. However, we assume only that $h(t) > 0$ for all $t$, allowing $h(t)$ to both increase and decrease over time, so that such changes are estimated from the data. For simplicity we assume that $h$ is piecewise constant on pre-defined subintervals of the modelled time interval $[0, T]$ with $h(0) = 1$ for identifiability, then $h(t) = h_{i-1}$ on the $i$'th subinterval ($2 \leq i \leq H$).

We now define the susceptibility $s_i$ and infectivity $t_i$ of a patch $i$ in terms of the covariates for that patch. There are many ways of defining the relationship between susceptibility/infectivity and the covariates. Here we suggest a general approach. Typically, covariates are either compositional or non-compositional [28]. Non-compositional covariates have unconstrained values, for example, average temperature and rainfall for the patch. Compositional covariates are constrained and dependent, for example, the proportion of the patch occupied by different land uses. These covariates are proportions that sum to one. Suppose that the set of covariates $\mathcal{C}_i$ for patch $i$ comprises non-compositional covariates $c_{i,k}, 1 \leq k \leq K$ and compositional covariates $h_{i,l}, 1 \leq l \leq L$, with $h_{il}$ being the proportion of patch $i$ on which a spatial categorical variable takes value $l$ (of $L$ distinct possible values). For example, these proportions may be obtained from a partition of each patch into $L$ distinct land uses. Then, a plausible model for the susceptibility of a patch is $s_i = (\sum_{l=1}^{L} h_{i,l}\sigma_l)exp(\sum_{k=1}^{K} c_{i,k}\sigma'_k)$ where the Greek letters are parameters to be estimated and each $\sigma_l \geq 0$. We also require that $\sum_{l=1}^{L} \sigma_l = 1$ for identifiability, bearing in mind that $s_i$ will be multiplied by the overall rate $\rho$ in the expression for the secondary force of infection. This general form of susceptibility function is analogous to the suitability function defined in [27] in the context of biological invasions. A similar expression is assumed for patch infectivity $t_i = (\sum_{l=1}^{L} h_{i,l}\gamma_l)exp(\sum_{k=1}^{K} c_{i,k}\gamma'_k)$. Again, the Greek letters $\gamma'_k$ for $1 \leq k \leq K$ and $\gamma_l$ for $1 \leq l \leq L$ are parameters to be estimated, with each $\gamma_l \geq 0$ and $\sum_{l=1}^{L} \gamma_l = 1$.

Each of the $L$ categories contributes additively to the susceptibility, with the contributions weighted by the covariates $h_{i,l}$. Each non-compositional covariate contributes multiplicatively, with negative relationships possible for these covariates when $\sigma'_k < 0$. However, it is important to note that negative relationships are also possible for the compositional covariates, even though the $\sigma_l$ are non-negative, due to the constraint that $\sum_{l=1}^{L} h_{i,l} = 1$. For example, if the smallest component of $\sigma$ is $\sigma_q$ then the covariate $h_{i,q}$ will be negatively related to susceptibility in the sense that, if $h_{i,q}$ increases and the $h_{i,l}$ for $l \neq q$ decrease subject to the summation constraint, then the overall effect is that $\sum_{l=1}^{L} h_{i,l}\sigma_l = h_{i,q}\sigma_q + \sum_{l \neq q} h_{i,l}\sigma_l$ will decrease.

## Bayesian inference and prediction based on disease case reports

This section describes details of the approach to estimating model parameters from data, how we compare between different model formulations, quantify uncertainty in model predictions and test performance of the iPAR framework. Some readers may wish to skip these details on initial reading.

**Parameter and latent variable estimation.** Having defined the model, we now present methodology that allows fitting of the model to data including estimation of unobserved (latent) infection times. For a disease outbreak, case

reports describe observed cases of the disease in individual animals or on individual farms. We assume that each case has a location and a detection time associated with it. Locations are taken to be specified at patch resolution or better. Detection times of disease cases are taken to be interval-censored, which allows for the common scenario whereby new case reports are provided at regular intervals (e.g., daily, or monthly), but can also accommodate irregular observation times. We denote these reporting times by $0 = t_0 < t_1 < \ldots < t_N = T$ where $[0, T]$ is the time interval over which case report data are available. We assume that interval-censored detection times should constrain the actual infection times of the individuals/farms within the patch, i.e., that a case occurring after time $t_i$ but before time $t_{i+1}$ is reported at time $t_{i+1}$. In practice, we expect performance of this modelling framework to degrade if a significant number of infections occurring in an interval $(t_i, t_{i+1})$ are unobserved at $t_{i+1}$. To some extent this can be countered by extending these reporting time windows used in the analysis resulting in greater estimated uncertainty in inferred infection times (see below). When aggregating to the patch level, we define the infection time of a patch to be the earliest recorded infection time of the set of individuals/farms within it. We assume that all infected patches are detected; in this sense the system is taken to be perfectly observed, a common assumption in the literature, see, e.g., [19,29].

We use Bayesian techniques to infer the model parameters from outbreak data. The true infection time of each patch is not known with certainty so, as is common when fitting epidemic models, we use data augmented MCMC to facilitate the inference [7]. This general approach to inference is well established in the literature and has been applied in, for example, [29–31]. Let $\theta$ denote the set of model parameters and let $\mathcal{U} = \{U_i; 1 \leq i \leq I\}$ denote the set of unobserved patch infection times. For each patch $i$, let $T_i \epsilon \{0, 1, 2, \ldots, N+1\}$ denote the 'time category' into which the patch falls in terms of reporting. So, $T_i = 0$ if $i$ was already reported as infected at time $t_0 = 0$ and $T_i = j$ ($1 \leq j \leq N$) if $i$ was first reported as infected at reporting time $t_j$. Finally, $T_i = N+1$ if $i$ was not reported as infected at any point in the time interval $[0, T]$, which we assume implies that $i$ remains susceptible at time $T$. Finally, let $\mathcal{T}$ denote the set of observed $T_i$. These observed time categories $\mathcal{T}$ are the data we use to infer the model parameters $\theta$ and the actual and unobserved or latent patch infection times $\mathcal{U}$. Bayes' theorem states that

$$P(\theta, \mathcal{U}|\mathcal{T}) \propto P(\mathcal{T}, \mathcal{U}|\theta) P(\theta)$$

Here $P(\theta)$ represents the joint prior distribution of the model parameters. The prior distribution is typically chosen to be non-informative (see S2 Appendix for details). The factor $P(\mathcal{T}, \mathcal{U}|\theta)$ above is usually termed the complete data likelihood for the model parameters conditioned on the observed data and a given set of latent infection times. The transmission process is a continuous time discrete state space Markov process, so this likelihood can be computed using standard techniques as described in, for example, [30,32]. We are now in a position to draw samples from the joint posterior distribution $P(\theta, \mathcal{U}|\mathcal{T})$. This is achieved by Markov Chain Monte Carlo (MCMC) methods. Full details of the likelihood calculation and sampling algorithm may be found in S3 Appendix. The computer code for running the simulations and inference was written in C, with R used for pre- and post-processing of data.

**Model selection.** We employ a widely used metric for assessing goodness of fit, the Deviance Information Criterion (DIC) [33]. The DIC is not uniquely defined for latent variable models [34]. In [14] they assessed a number of DIC variants in the context of spatio-temporal epidemic modelling and found that their metric $DIC_1$ (which corresponds to $DIC_6$ in [34]) performed well. Therefore, we use this metric to compare models, in particular to compare the constant-in-time and varying-in-time models when both fitted to the same set of outbreak data. See S4 Appendix for details of the definition and computation of this metric. Model selection is a challenging problem, and is not the main focus of this manuscript.

**Uncertainty in model predictions.** Having fitted the model to outbreak data, we can then use the fitted model to predict the timing of future transmission events beyond the period of the observed data by sampling from the appropriate posterior predictive distribution. This is achieved by forward simulation of the transmission model over a time interval $[T, T+S]$, with model parameters sampled from the joint posterior distribution and initial conditions typically set to the

infection status of the patches at the final time point of the data used to fit the model ($t = T$). We typically conduct large numbers of such simulations, at least 10000 say, so as to capture the full distribution of possible outcomes.

The model predictions obtained from sampling the posterior predictive distribution can be summarised in a number of ways. We can plot the number of infected patches as a function of time. We can also plot risk maps that depict, for each patch $i$, the probability that it is infected by some future time $t = T + S$, which we calculate as the proportion of simulations in which $i$ becomes infected by time $t = T + S$.

The use of the constant-in-time model to predict future events in this way is straightforward. On the other hand, fitting the varying-in-time model leads to an estimate of the function $h(t)$ for $t < T$, but we cannot use this fitted model to make predictions without further assumptions on the behaviour of $h(t)$ for $t > T$. For this reason, we only use the constant-in-time model to make predictions, although one could envisage future projections conditioned on specified scenarios for $h(t)$ at $t > T$.

The predictions can also be compared to what actually happened, if such data are available. This is possible for simulated data where we generate a realisation of the model that represents the 'truth'. Alternatively, with real data from a disease outbreak we can hold out some portion of the data from later stages of the outbreak, fitting the model only to data from the earlier stages of the outbreak. In either case, we can compare the predictions to such 'truth' in two ways. First, we may directly compare the *total number* of infected patches as it increases over time. Second, we can assess the ability of the predictions to identify the *correct* infected patches, rather than simply the correct overall number of infected patches, by computing *discrimination* metrics that compare the risk map to the true map of infection. For a given probability threshold $p$, we calculate the true positive rate (TPR, the proportion of patches infected during $[T, T + S]$ with infection probability exceeding $p$) and the false positive rate (FPR, the proportion of patches uninfected at time $t = T + S$ with infection probability exceeding $p$). Plotting the true and false positive rates for multiple thresholds $p$ gives the Receiver Operating Characteristic (ROC) curve [35], from which standard metrics such as the AUC (Area under the ROC curve) and the TPR at 5% FPR can be derived.

**Testing the iPAR modelling approach.** To assess the iPAR framework, it is important to first test it using simulated outbreak data. The use of simulated data has two key benefits. First, we can explore a wider range of scenarios than is possible using real outbreak data, which can be very helpful when exploring the strengths and weaknesses of methods, as well as the limits of their applicability. Second, with simulated data we always know the 'true' data generating process, which allows us to assess, for example, how close parameter estimates obtained from the simulated data are to their 'true' values. Such comparisons between true and estimated parameter values are impossible with real outbreak data. On the other hand, simulated data may have properties that are different from real outbreak data, and the conclusions may depend on the choice of synthetic trajectories. Therefore, it is also critical to test methods via application to real world data and problems as we do in Case study. Model assumptions can be validated externally by comparing fitted model outputs with other sources of data on the outbreak and the susceptible population that were not used to fit the model. Such approaches depend very much on the details of the scenario under investigation. We apply the iPAR framework to real-world disease case report data and describe some specific validation approaches in Case study.

Simulation-based methods are used to systematically test the constant-in-time iPAR model in Results. We employ a form of sensitivity analysis in which we start from a baseline model parameterisation based on the parameter estimates obtained from the ASF outbreak data in Case study. We then vary each model parameter separately (keeping all others fixed), simulating a single set of outbreak data for each parameter combination and then inferring model parameters from the simulated outbreak data. In this way, we can build up plots of true versus inferred parameter values (95% credible intervals) for each model parameter. This kind of sensitivity analysis has been conducted in, e.g., [36]. If the model parameters are sampled from the prior then the *average* coverage is equal to the nominal value [37] but, for a specific choice of parameters, Bayesian credible intervals don't necessarily have their nominal coverage. This means that the true value of a parameter may not fall within a 95% Bayesian credible interval for 95% of the time. Nevertheless, we can visually assess

the output from the sensitivity analysis in order to check (a) whether the inference is recovering the 'true' parameter values in the way we would expect, and (b) which parameters are more difficult to recover from the simulated data. For parameters that are difficult to recover, the inferred parameter may only depend weakly on the true parameter, and may instead be strongly influenced by the prior distribution assigned to the parameter.

In the setting of the constant-in-time model with no non-compositional covariates, which is the setting we use in Estimation of key epidemiological parameters in Results, the full set of model parameters is as follows: the land use susceptibilities $\sigma_1, \ldots, \sigma_L$, the land use infectivities $\gamma_1, \ldots, \gamma_L$, and three remaining parameters $\lambda$ (kernel parameter), $\rho$ (overall transmission rate) and $\varepsilon$ (background transmission rate). In order to perform the sensitivity analysis, each parameter is taken in turn and varied while keeping other parameters fixed. One issue is that the susceptibilities are not independent, being constrained to sum to 1, and similarly for the infectivities. To circumvent this problem, we reparametrize the model using the isometric logratio transform $\xi = \mathsf{ilr}(\sigma) \in \mathbb{R}^{L-1}$, $\eta = \mathsf{ilr}() \in \mathbb{R}^{L-1}$, which maps the simplex-valued parameters to real multivariate parameters (see S5 Appendix for details). Thus, the components of $\xi$ and $\eta$ are unconstrained and can be varied independently as part of the sensitivity analysis.

## Results

### Assessing the iPAR framework performance

We now assess the iPAR framework by applying it to simulated outbreak data. In Estimation and prediction for an illustrative simulated outbreak, we analyse data from a single simulated outbreak – simulated from the constant-in-time model with spatially varying susceptibility/infectivity – in order to illustrate the typical workflow and outputs obtained from the model fitting process. In Estimation of key epidemiological parameters, we conduct a systematic simulation study in which we generate outbreak data for a wider range of scenarios, though still for the constant-in-time model with spatially varying susceptibility/infectivity, and assess the ability of the inference to recover the 'true' parameters. In Estimation of temporal trends in transmission, we explore the ability of the inference to recover temporal trends in transmission, using simulated outbreak data from the varying-in-time model with spatially varying susceptibility/infectivity. Finally, in Benefits of modelling spatial variation in susceptibility and infectivity, we assess the benefits of modelling spatial variation in susceptibility and infectivity, one of the key aspects of the iPAR framework, both in terms of estimating key parameters and in terms of predicting held-out future simulated outbreak data. To do this, we generate simulated outbreak data from the constant-in-time model with spatially varying susceptibility/infectivity, but we fit either a model of similar form or a model which assumes spatially homogeneous susceptibility/infectivity.

The results of the assessments carried out here demonstrate that the iPAR framework is able to reliably infer epidemiologically relevant information from case-only reports across the range of scenarios considered. We found that the simulations run very quickly but the inference code can take between 1 and 5 hours to run for the scenarios presented here. The time taken increases with the number of patches and the number of patches that become infected, but in a non-straightforward manner due to optimisations implemented in the inference code. Computations were carried out using an Intel i7-8665u processor with 32GB memory. Five MCMC chains were run in parallel, with 100000 iterations per chain. Simulated trajectories were selected at random. Some readers may wish to skip the following details on initial reading and proceed directly to Case study, later returning to gain greater insights into those latter results.

### Estimation and prediction for an illustrative simulated outbreak

Outbreak data are simulated using the constant-in-time iPAR model on the Estonian landscape, which is divided into 575 10km square patches, with model parameters similar to those estimated from the outbreak data for ASF in wild boar in Estonia (see Case study). As in the case study, land use data for Estonia are used as the basis for computing covariates used to model spatial variation in both susceptibility and infectivity. Four patches are initially seeded with infection, and

the duration of the simulation is chosen such that a substantial proportion of patches become infected by the end of the simulation.

A single simulated outbreak can be summarised by plotting the number of infected patches over time (Fig 1A, black line). The constant-in-time iPAR model was fitted to the early-stage epidemic data up to month 12 from this single simulated outbreak, with simulated patch infection times interval censored into one-month long intervals. The resultant posterior distribution provides estimates for all model parameters. For example, we can visualise the posterior marginal distribution of the background transmission rate parameter $\varepsilon$ (Fig 1B, with true value represented by a vertical line) by aggregating MCMC samples. We can also generate the 95% credible intervals for parameters and functions of the model parameters such as the transmission kernel (Fig 1C, true kernel in black, 95% credible interval in blue shading). Credible intervals for the components of the susceptibility parameter $\sigma$ (Fig 1D, 95% credible intervals represented by blue lines) are often quite wide, but in general these intervals align well with the 'true' values.

The model, fitted to data up to month 12, was then used to predict progression of the epidemic beyond the first 12 months. This was achieved by simulating 10000 times with parameters sampled from the posterior distribution and initial conditions specified by the state of the epidemic at month 12. These predictions may be summarised by 95% credible intervals for the number of infected patches over time (Fig 1A, blue shading). The intervals contain the 'true' epidemic curve in this case, suggesting satisfactory inference of the model parameters.

### Estimation of key epidemiological parameters

We systematically tested the constant-in-time model over a wide range of parameter combinations using the sensitivity analysis approach described in Testing the iPAR modelling approach in Methods. The simulation model set-up was identical to that in the previous section except for the use of different sets of model parameters. The data from most of the simulated outbreak (up to month 52) was used to fit the model. Plots of true versus inferred parameters are available in Fig 2 for $\lambda$, $\rho$, $\varepsilon$ and in S6 Appendix for the remaining parameters. Key model parameters such as the kernel parameter $\lambda$ and the background transmission rate $\varepsilon$ are well recovered from the simulated outbreak data, though the overall transmission rate $\rho$ tends to be somewhat overestimated. Reasons for the latter phenomenon will be discussed later. The susceptibility parameters tend to be well recovered, while the infectivity parameters seem more difficult to recover, which is consistent with the results in [36].

### Estimation of temporal trends in transmission

Next, we focus on the varying-in-time iPAR model with parameters similar to those obtained from fitting this variant of the model to the Estonian ASF outbreak data (see Case study) as a baseline parameterisation. We then vary the form of the temporal trend function $h$ according to three pre-chosen scenarios: (1) $h$ decreasing linearly (perhaps representing increasing levels of disease control), (2) $h$ constant (no disease control), and (3) $h$ initially constant then suddenly drops to low levels (corresponding to rapid strong disease control after some delay). The changepoints of the function are fixed and are kept equal to those used in the case study. The function $h$ from the first scenario is closest to what was actually estimated from the data in the case study. Note that, in the case study, $h$ was seen to initially increase before decreasing, which is why we have also allowed for an initial increase in $h$ in the first and third scenarios. We assess the extent to which these temporal trends can be recovered by inference from data simulated under each of the above three scenarios. We also fit the constant-in-time model to the same sets of simulated outbreak data, which allows us to compare model fits (constant-in-time versus varying-in-time) using $DIC_1$ as discussed in Model selection in Methods.

Fig 3 shows the simulated 'true' incidence curve for each of the three temporal trends. Overlaid on this 'true' incidence curve is a region representing 95% credible intervals for incidence, obtained from simulations of the fitted varying-in-time model with initial conditions the same as those of the simulated outbreak data used for fitting the model. For all three

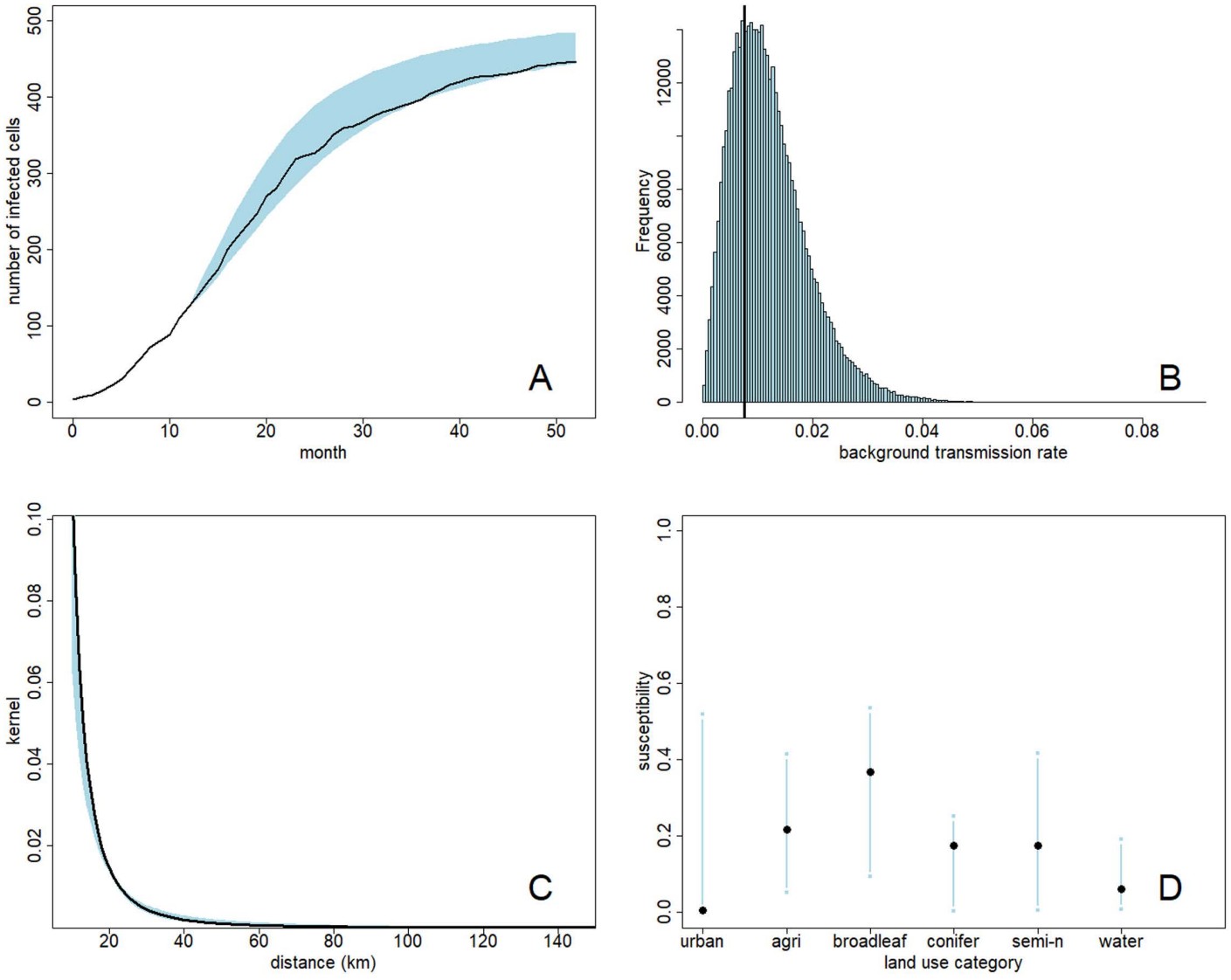

**Fig 1. Estimated parameters and predictions from fitting the iPAR model to simulated outbreak data.** A: number of infected patches over time in the simulated outbreak data (black line). The model was fitted to the outbreak data up to month 12. Predicted number of infected patches (95% credible interval) beyond month 12 is shown in blue. B: marginal posterior distribution for the background transmission rate, with 'true' value indicated by the black vertical line. C: 'true' kernel function (black line) and estimated kernel function 95% credible interval (blue shading). D: estimated susceptibility parameter $\sigma$ for each land use (95% credible interval, blue) and 'true' value (black circle).

temporal trends, the agreement between the credible intervals and the 'true' incidence curve shows that the algorithm is able to reliably recover parameters giving temporal trends in incidence that agree with the 'true' incidence curve. Fig 3 also shows the 95% credible interval for $\rho h(t)$ in the same set of three simulations. Despite the striking fact that the form of this function is not at all obvious if presented with a single outbreak incidence curve, the trend function $\rho h(t)$ is inferred correctly from single outbreaks, though with some overestimation. We also computed $DIC_1$ values for each model fit (see S4 Appendix for actual values obtained). For scenarios (1) and (3), in which $h \neq 1$, the $DIC_1$ indicated a strong preference for the varying-in-time model. For scenario (2), in which $h = 1$, there was a mild preference for the constant-in-time model. These results were all as we would expect.

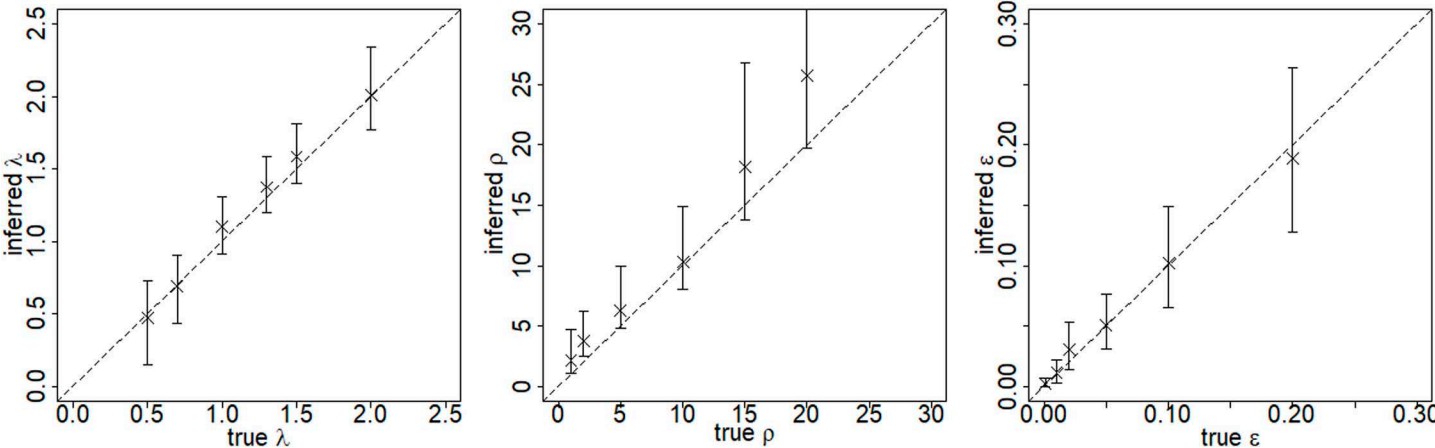

**Fig 2. Reliable parameter inference for the constant-in-time iPAR model.** Each panel corresponds to a parameter of the model. The 'true' value of the parameter is plotted against the posterior median (cross) and posterior 95% credible interval (vertical line). For comparison purposes we also super-impose the diagonal line representing perfect agreement between inferred and true values. This figure includes panels for the kernel parameter $\lambda$, the overall transmission rate $\rho$ and the background transmission rate $\varepsilon$. Analogous figures for the other parameters may be found in S6 Appendix.

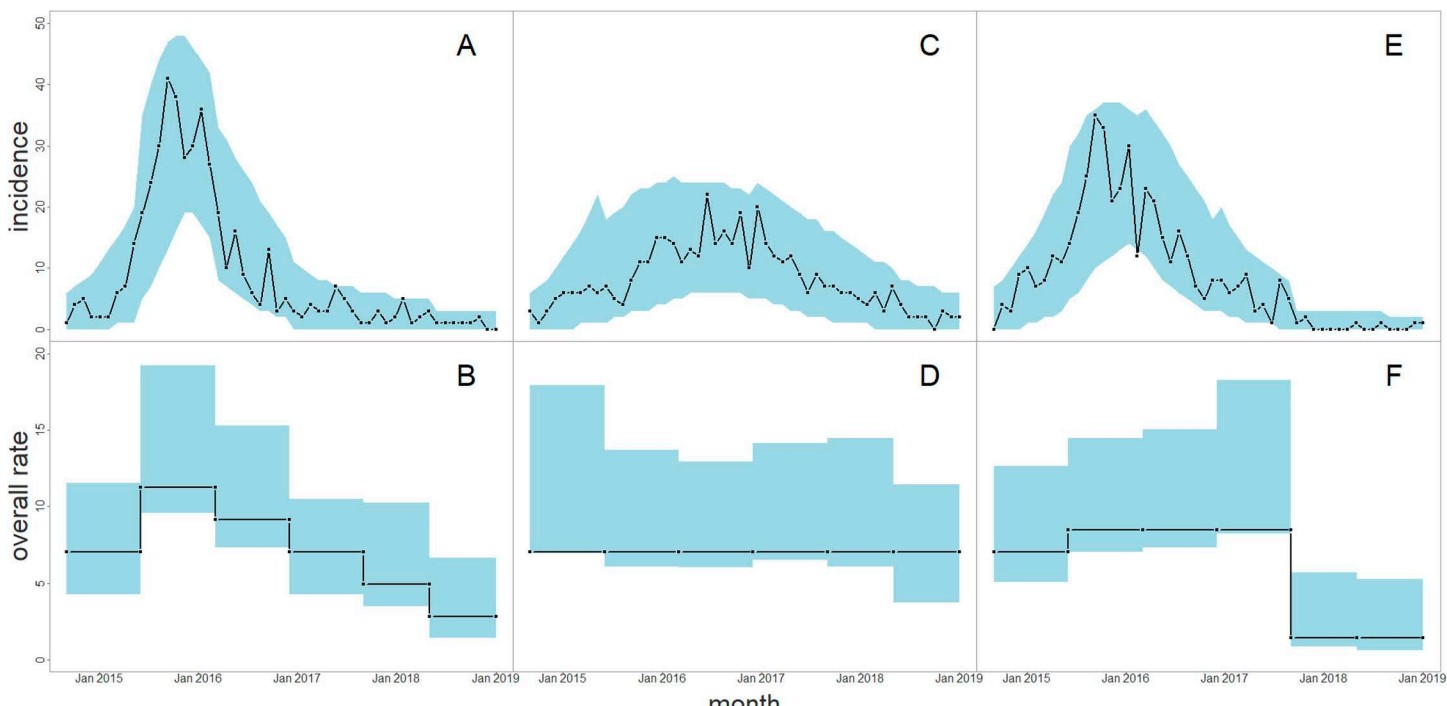

**Fig 3. Reliable recovery of temporal trends in the time varying simulation study.** A: the black line shows the incidence curve from a single simu-lation of the "linear decrease" scenario. The blue area shows the 95% credible intervals for incidence, defined as newly infected patches each month, obtained from the posterior predictive distribution of the fitted varying-in-time model, given the data from that single simulation. B: true and estimated $\rho h(t)$ for the same scenario. Black line represents true value, blue area shows the 95% credible interval. C,D: same plots for the "constant over time" scenario. E,F: same plots for the "sudden drop off" scenario (see text for details).

## Benefits of modelling spatial variation in susceptibility and infectivity

We assess the importance of modelling spatial variation in susceptibility and infectivity in the context of the constant-in-time iPAR model with land use covariates but no non-compositional covariates. The spatial variation in the patch susceptibilities and infectivities of the model arises as a result of spatial variation in the covariates, combined with differences in the susceptibility and infectivity parameters between land use categories. Our approach here is to simulate outbreak data incorporating such spatial variation and then fit two model variants to the simulated outbreak data. First, we fit a heterogenous model which accounts for this spatial variation and has the same form as the model used for simulation. Second, we fit a simpler homogeneous model in which there are no covariates and patch susceptibility/infectivity is spatially constant. Comparing the two model fits and their associated predictions enables us to assess the importance of taking into account variation in susceptibility and infectivity.

We consider model fitting given two levels of data availability. First, we fit models to data from the early stages of an outbreak. Second, we fit models to data from an entire outbreak. At both levels, we consider four different parameter combinations for the simulated outbreak data. This is less than the number of parameter combinations considered in Estimation of key epidemiological parameters, however the chosen parameter combinations cover a range of common transmission patterns. We use either a wide or a moderately wide transmission kernel. Narrow kernels, such as those approximating nearest neighbour kernels, are avoided because we intuitively expect there to be little difference between the predictive performance of the heterogenous and homogeneous models in this case – see Case study and associated discussion for more on this topic. In the simulations, we assume that there is a single land use with high susceptibility (all other land uses are assigned equal low susceptibilities), and the same applies for infectivity. Broadleaf woodland is taken to have high susceptibility in all scenarios, while the high infectivity land use alternates between broadleaf woodland and arable land. Precise details of the scenarios are provided in S8 Appendix.

We repeat the model fitting and prediction for 20 replicate simulations per parameter combination, in order to give a fuller picture of parameter recovery and predictive performance in each case. Mean (over all 20 replicates) AUC and mean TPR at 5% FPR measure discriminatory power. We compute the posterior predictive distribution of $N_{est}/N_{truth}$, the predicted number of infected patches divided by the true number of infected patches, and its reliability which is the % of replicates in which the 95% CrI for $N_{est}/N_{truth}$ includes $N_{est}/N_{truth} = 1$. We also compute the mean bias (posterior median of $N_{est}/N_{truth}$ minus 1, averaged over all 20 replicates). Corresponding quantities are computed for the kernel parameter $\lambda$. For the susceptibility and infectivity parameters we adopt a different approach. We focus on estimation of the large susceptibility parameter and the large infectivity parameter. These parameters are likely more difficult to estimate than the other susceptibility/infectivity parameters because they are far from the main mass of the prior distribution on the simplex. So, we measure the difference between the posterior and prior medians (averaged across all 20 replicates), which captures the extent that the posterior has moved away from the prior. If the difference is very small then there is little information in the data regarding that parameter. For the given simulation set up, a posterior median that is 0.73 higher than the prior median corresponds to perfect estimation of the large susceptibility/infectivity parameter.

Using only early-stage data, predictive performance assessed using discrimination metrics was consistently better for the heterogeneous model, if not by a very large margin (Table 1). Ability to predict $N_{est}/N_{truth}$ was good for both the heterogenous and homogeneous models. In terms of parameter recovery using only early stages data, we focus on the transmission kernel parameter, the susceptibility parameter associated with the high susceptibility land use class and the infectivity parameter associated with the high infectivity land use class. The kernel parameter was typically well estimated (Table 2), though the homogeneous model tended to somewhat overestimate this parameter, a phenomenon that was also observed by [27]. However, susceptibility and infectivity parameters were imperfectly recovered (Table 2). Comparison of the posterior and prior medians suggests that the early-stage data used for fitting provides some information on susceptibility but is less informative on infectivity.

**Table 1. Early-stage data predictive performance.**

| Scenario | | | | Discrimination metrics | | $N_{est}/N_{truth}$ | |
|---|---|---|---|---|---|---|---|
| Kernel | Susc | Infe | Model | mean AUC | mean TPR @ 5% FPR | reliability (%) | mean bias |
| wide | broadleaf | broadleaf | homog | 0.67 | 0.13 | 85 | -0.02 |
| | | | heter | 0.71 | 0.18 | 90 | -0.03 |
| wide | broadleaf | arable | homog | 0.70 | 0.15 | 95 | 0.00 |
| | | | heter | 0.74 | 0.20 | 95 | -0.01 |
| moderate | broadleaf | broadleaf | homog | 0.74 | 0.18 | 95 | 0.02 |
| | | | heter | 0.78 | 0.25 | 100 | -0.00 |
| moderate | broadleaf | arable | homog | 0.74 | 0.19 | 85 | 0.04 |
| | | | heter | 0.78 | 0.26 | 90 | 0.01 |

Models are fitted to data on the early stages of simulated outbreaks on the Estonian landscape. Twenty independent replicates were carried out for each of the four parameter combinations. For each parameter combination (choice of transmission kernel and choice of high susceptibility and high infectivity land uses), we compare predictions from fitted homogeneous and heterogeneous models at four months beyond the final month of the simulated outbreak data used to fit the model.

**Table 2. Early-stage data parameter recovery.**

| Scenario | | | | Kernel parameter | | Susceptibility | Infectivity |
|---|---|---|---|---|---|---|---|
| Kernel | Susc | Infe | Model | reliability (%) | mean bias | mean post-prior | mean post-prior |
| wide | broadleaf | broadleaf | homog | 95 | 0.04 | | |
| | | | heter | 95 | 0.02 | +0.29 | -0.05 |
| wide | broadleaf | arable | homog | 100 | 0.09 | | |
| | | | heter | 100 | 0.02 | +0.37 | -0.03 |
| moderate | broadleaf | broadleaf | homog | 90 | 0.07 | | |
| | | | heter | 100 | 0.02 | +0.43 | +0.01 |
| moderate | broadleaf | arable | homog | 70 | 0.10 | | |
| | | | heter | 80 | 0.04 | +0.45 | +0.07 |

Models are fitted to data on the early stages of simulated outbreaks on the Estonian landscape. Twenty independent replicates were carried out for each of the four parameter combinations. Mean post-prior is the mean, over replicates, of the posterior median minus the prior median for the parameter.

In the 'entire outbreak' data availability setting, it is no longer meaningful to consider predictive performance on held-out future outbreak data. Parameter recovery is assessed for the same set of parameter combinations as for the 'early stages' setting. Reliability of inference for the transmission kernel parameter is excellent for the heterogenous model but less good for the homogenous model (Table 3). This parameter is consistently overestimated but especially by the homogeneous model, which again is consistent with what was observed in [27]. The posterior medians of the (large) susceptibility and infectivity parameters have shifted closer to their true values when compared to Table 2. This is as we would expect given that we have more data available to fit the model in this setting. In particular, there is now clearer evidence of a shift in the infectivity parameter posterior distribution away from the prior distribution.

Finally, to put these results into context, we take a closer look at the spatial variation in the Estonian landscape, which we have used for these simulations. At the 10km resolution, the Estonian landscape is not particularly heterogeneous in terms of land use (Fig 4). Most of the 575 10km square patches comprise a mixture of land uses, with very few patches consisting mostly of a single land use. The low heterogeneity in land use is one factor leading to limited variation in estimated patch susceptibilities (Fig 5). In general, we might expect that increased heterogeneity in land use, increased

**Table 3. Full outbreak parameter recovery.**

| Scenario | | | | Kernel parameter | | Susceptibility | Infectivity |
|---|---|---|---|---|---|---|---|
| Kernel | Susc | Infe | Model | reliability (%) | mean bias | mean post-prior | mean post-prior |
| wide | broadleaf | broadleaf | homog | 75 | 0.19 | | |
| | | | heter | 95 | 0.05 | +0.64 | +0.05 |
| wide | broadleaf | arable | homog | 65 | 0.20 | | |
| | | | heter | 95 | 0.08 | +0.58 | +0.05 |
| moderate | broadleaf | broadleaf | homog | 75 | 0.13 | | |
| | | | heter | 90 | 0.03 | +0.63 | +0.12 |
| moderate | broadleaf | arable | homog | 70 | 0.16 | | |
| | | | heter | 95 | 0.04 | +0.61 | +0.15 |

Models are fitted to data on entire simulated outbreaks on the Estonian landscape. For other details, see text for Table 2. Mean post-prior is the mean, over replicates, of the posterior median minus the prior median for the parameter.

differences in susceptibility/infectivity between land uses, a wide kernel and higher numbers of available case reports are all factors that should increase the benefits of modelling spatial variation in susceptibility and infectivity.

## Case study: African swine fever in the Baltic states

### Background/approach

Here, we used all ASF cases in wild boar populations reported in Estonia between 2014 and 2019 to the World Organisation for Animal Health (WOAH). The Estonian ASF outbreak had very serious impacts for both the pig industry and wildlife. African Swine Fever leads to high mortality, and its spread is difficult to control. It continues to pose a major ongoing threat

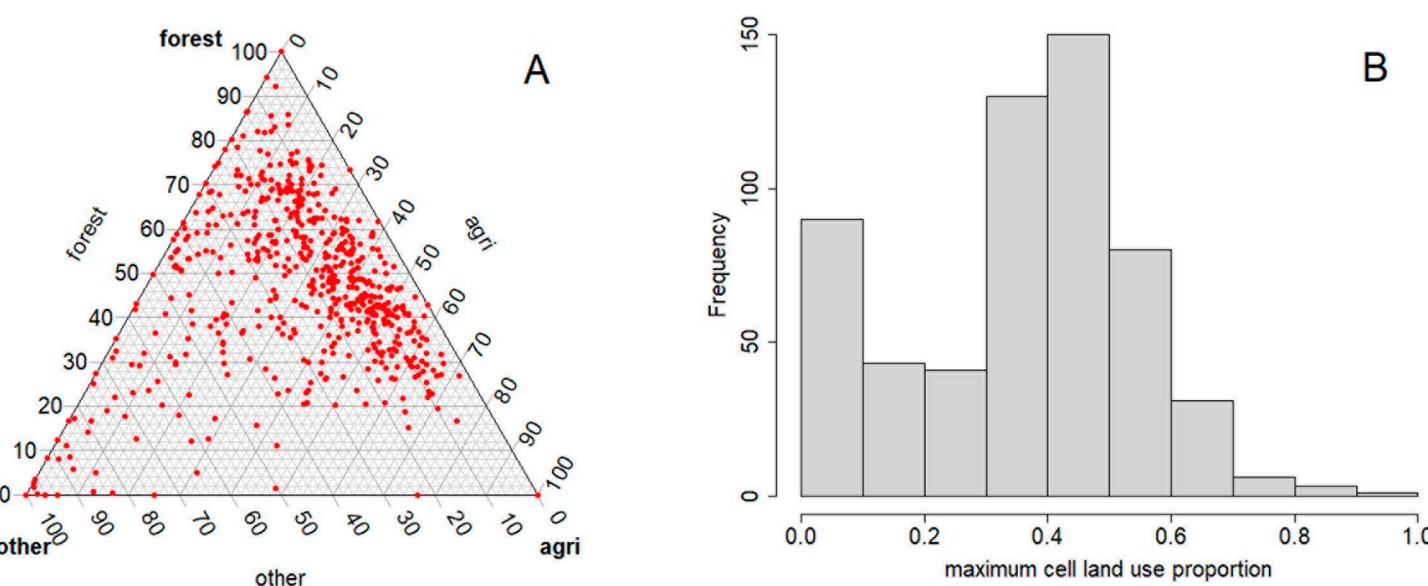

**Fig 4. Landscape heterogeneity in Estonia.** A: ternary diagram showing the land use composition of every 10km square patch covering Estonia (land use amalgamated into three categories – forest, agricultural and other – for ease of visualisation). B: histogram showing maximum patch land use proportion for every 10km square patch in Estonia, with six land uses as in the model covariates.

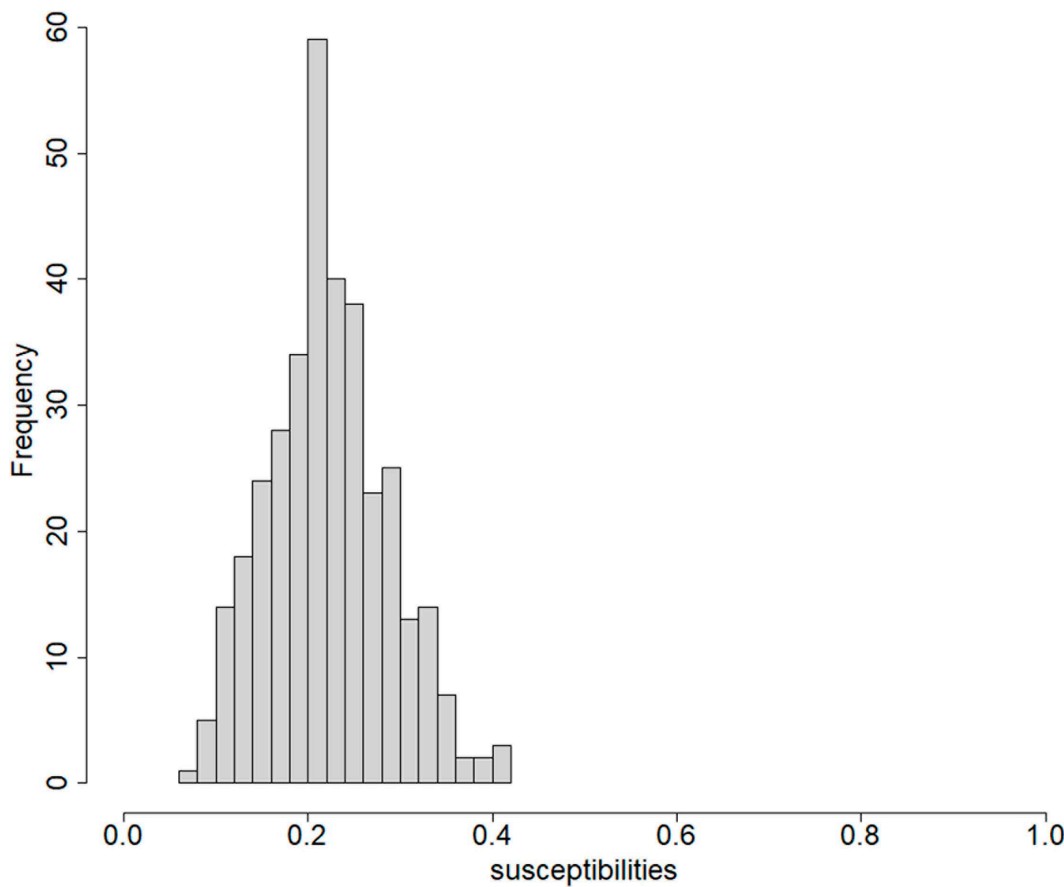

**Fig 5. Typical distribution of inferred patch susceptibilities.** Typical distribution of (inferred) posterior mean patch susceptibilities given data from a simulated disease outbreak in Estonia. There is some variation but few patches with very high or low susceptibilities.

in many regions of Europe [38–40]. In this study we focus on better quantifying risk of transmission and spread of ASF in the Estonia wild boar population. Knowledge of the wild boar population in this region is incomplete, though estimated wild boar density maps have been published [39]. Within Estonia, there have been extensive efforts to control the ASF outbreak, partly by reducing the density of wild boar. These efforts are reflected in density maps showing an apparent reduction in the population over time [39]. Case reports of ASF in dead wild boar are available at monthly time intervals, with detailed location information. Therefore, given the uncertainty in knowledge of the Estonian wild boar population described above, we apply the iPAR framework to this outbreak.

To apply the framework to the ASF outbreak in Estonian wild boar, we must first define a grid of patches covering Estonia, as well as covariates likely to be related to wild boar density and hence to patch susceptibility and infectivity. Land use is one of the key factors likely to determine wild boar density [41], and hence we use high resolution CORINE land use data available for Estonia [42]. The case report data for wild boar in Estonia also have high spatial resolution. However, if the patches used to define the infectious units of the model are too small then (a) fitting the model will be very computationally intensive, and (b) the persistence assumption inherent in our assumed SI framework is less likely to be reasonable. As a compromise we initially opted for square patches of size 10km by 10km, though smaller patches would also be feasible. This choice of patch size was informed by a persistence of infection analysis, which revealed that, at the 10km resolution, ASF in wild boar in Estonia typically persists for long periods of time, often multiple years (S9 Appendix). Any of

the 10km patches composed wholly of sea or inland water were removed from the set of patches. In the case of Estonia, there are 575 patches covering the country. Fig 6 illustrates the spatial spread of ASF through the wild boar population in Estonia at the 10km patch resolution.

We took CORINE land cover data [42] at 100 metre resolution and spatially aggregated it to the 10km patches to obtain proportions of each land use category for each patch. We also aggregated some of the CORINE land cover categories, resulting in a total of 6 categories: 1 = urban, 2 = agriculture, 3 = broadleaf/mixed forest, 4 = coniferous forest, 5 = semi-natural, 6 = wetlands/water bodies excluding sea. We defined the susceptibility of a patch $i$ to be $s_i = \sum_{k=1}^{6} h_{i,k}\sigma_k$

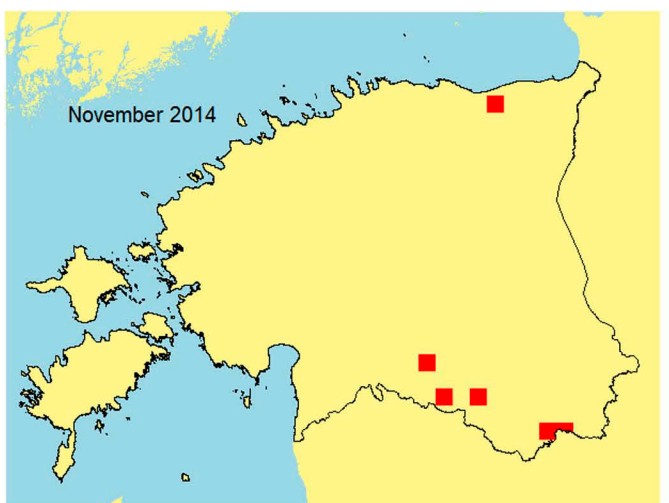
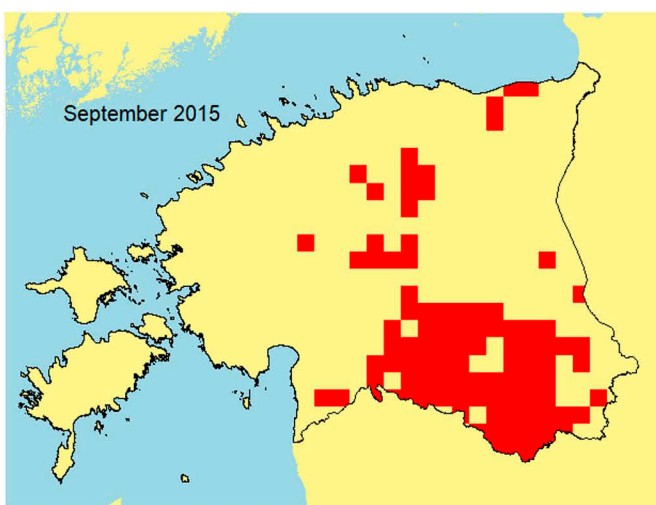
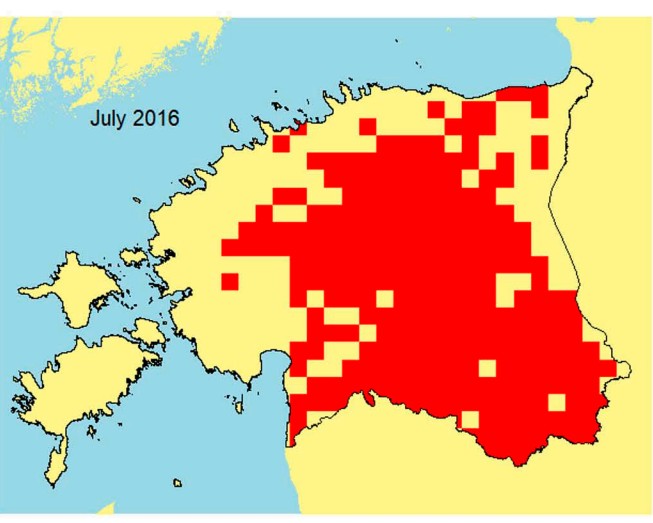
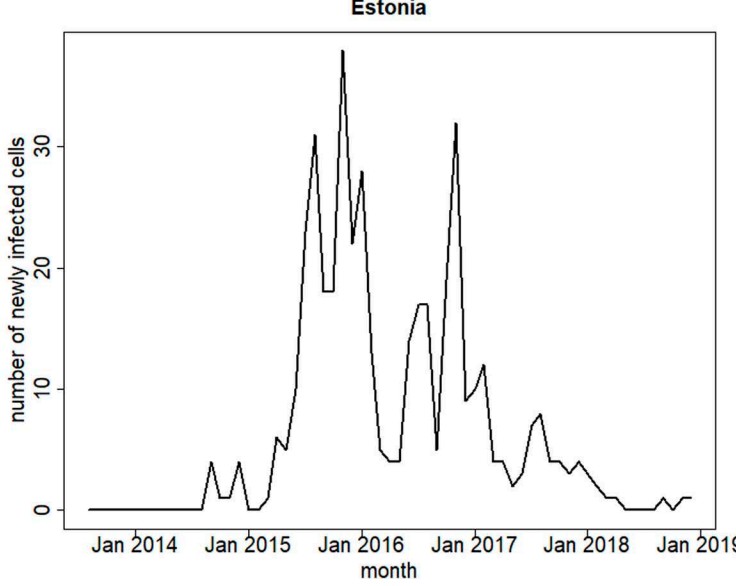

**Fig 6. Spread of ASF in Estonian wild boar.** Spatially aggregated African Swine Fever wild boar cases reported in Estonia. Snapshots at: November 2014, September 2015, July 2016. Bottom right panel shows the outbreak incidence curve, generated using spatially aggregated wild boar case reports. Map base layer taken from https://gadm.org/maps.html.

where the $\sigma_k$ are parameters to be estimated and $h_{i,k}$ is the proportion of land in patch $i$ occupied by land use category $k$. Similarly, consistent with the notation in Methods, the infectivity of a patch $i$ is defined as $t_i = \sum_{l=1}^{6} h_{i,l}\gamma_l$.

The latent period of ASF in wild boar is very short relative to the scale of the spreading process, so we have chosen to ignore this aspect of disease dynamics, which is implicit in the SI assumption. In addition, as discussed above the SI assumption represents the patch-scale dynamic which may differ from disease progression in an individual host. A case report from the outbreak is associated with a specific month. So, the time of the case report is considered to be interval-censored, with the time interval equal to the relevant month. The model can now be fitted to the outbreak data as described in Methods. We could have taken into account other covariates but, for simplicity, we chose to use land use data only. This case study is intended as a proof of principle, rather than a definitive analysis of ASF transmission in wild boar in Estonia.

## Parameter estimates

The constant-in-time and varying-in-time heterogeneous models were fitted to the full set of Estonian outbreak data, with the estimated parameters as in Table 4. The $DIC_1$ metric was computed for both model fits, and was found to strongly favour the varying-in-time model (details in S4 Appendix). This is not surprising given the strong decreasing trend in $h(t)$ that was identified (Table 4). Generally, parameter estimates were consistent between the two model variants. There is much uncertainty, but broadleaf woodland seems to have the highest susceptibility, while agricultural land has the highest

**Table 4. Parameter estimates obtained from fitting the iPAR model to data from the entire Estonian outbreak.**

| parameter | Estonia constant in time | | | Estonia varying in time | | |
|---|---|---|---|---|---|---|
| | lower | median | upper | lower | median | upper |
| $\sigma_{urban}$ | 0.00 | **0.01** | 0.06 | 0.00 | **0.02** | 0.10 |
| $\sigma_{agri}$ | 0.02 | **0.09** | 0.19 | 0.03 | **0.11** | 0.22 |
| $\sigma_{broadleaf}$ | 0.28 | **0.42** | 0.56 | 0.17 | **0.30** | 0.44 |
| $\sigma_{conifer}$ | 0.08 | **0.20** | 0.32 | 0.06 | **0.19** | 0.32 |
| $\sigma_{semi-n}$ | 0.04 | **0.20** | 0.37 | 0.10 | **0.28** | 0.46 |
| $\sigma_{wetlands}$ | 0.02 | **0.07** | 0.14 | 0.03 | **0.08** | 0.17 |
| $\gamma_{urban}$ | 0.00 | **0.07** | 0.30 | 0.00 | **0.07** | 0.31 |
| $\gamma_{agri}$ | 0.56 | **0.77** | 0.92 | 0.27 | **0.52** | 0.77 |
| $\gamma_{broadleaf}$ | 0.00 | **0.03** | 0.13 | 0.01 | **0.14** | 0.37 |
| $\gamma_{conifer}$ | 0.00 | **0.02** | 0.12 | 0.00 | **0.09** | 0.31 |
| $\gamma_{semi-n}$ | 0.00 | **0.02** | 0.13 | 0.00 | **0.05** | 0.22 |
| $\gamma_{wetlands}$ | 0.00 | **0.03** | 0.14 | 0.00 | **0.05** | 0.21 |
| $\lambda$ | 1.30 | **1.45** | 1.63 | 1.37 | **1.52** | 1.70 |
| $\rho$ | 3.71 | **4.54** | 6.03 | 4.18 | **7.05** | 11.80 |
| $1000\varepsilon$ | 1.07 | **7.65** | 18.42 | 0.25 | **3.84** | 11.55 |
| $h_1$ | | | | 1.11 | **1.71** | 2.74 |
| $h_2$ | | | | 0.63 | **1.00** | 1.64 |
| $h_3$ | | | | 0.28 | **0.47** | 0.80 |
| $h_4$ | | | | 0.12 | **0.23** | 0.44 |
| $h_5$ | | | | 0.02 | **0.06** | 0.16 |

Posterior median and upper/lower limits of 95% credible intervals. The parameters $h_0 = 1, h_1, \ldots, h_5$ represent the piecewise-constant time-varying function $h(t)$ as described in Methods. The time intervals over which the function is constant are pre-specified and are as depicted in Fig 7E.

infectivity. External transmission occurs at very low levels, and the estimated kernel is relatively narrow. The estimates of the overall rate $\rho$ for the two models are not easily comparable because, in the varying-in-time model, $\rho$ is multiplied by the function $h(t)$ which is estimated as less than one for most of the observation time interval. So it is not surprising that the estimated value of $\rho$ is higher for the varying-in-time model.

## Model predictions

Finally, we assess the predictive performance of the model fitted to the Estonian data. We fit the constant-in-time model to the early stages of the outbreak in Estonia, holding out data from later stages of the outbreak in order to test the model's predictions. Both the heterogeneous and homogeneous variants of the model were fitted.

The predicted number of infected patches under the heterogeneous and homogeneous models tends to be overestimated, though the predictions of the heterogeneous model lie a little closer to the actual data (Fig 7A). The predicted number of infected patches from simulation of a heterogeneous varying-in-time model fitted to the entire outbreak is much closer to the actual data (Fig 7B). This indicates that the varying-in-time model is sufficiently flexible to capture the key trends in the observed cumulative incidence curve in Estonia.

## External validation

Many aspects of the fitted model are difficult to validate externally. However, one aspect of interest is the time varying function $h(t)$, which was estimated to initially increase from the baseline value of 1 in 2015 to a maximum of 1.71 in 2016, before dropping to 0.06 by the end of the outbreak (Table 4). The function is intended to model the effects of disease control measures, and in Estonia a key control measure was reduction of the wild boar population. The trend seen in the estimated time varying function is broadly consistent with reported trends in wild boar population density across Estonia based on information from hunters (see Fig 5 in [40]). This comparison provides some evidence for the validity of the fitted varying-in-time model (see Fig 7C–E).

## Quantitative risk assessment for disease control

In addition to the retrospective analysis described above which shows the impact of the control measures that were adopted, the iPAR framework also provides information relevant to the within-outbreak targeting of disease control activities. For example, Fig 8A illustrates the estimated susceptibilities of patches in Estonia for the constant-in-time model, identifying which locations are more likely to be susceptible to disease incursion. Although there are some areas with greater susceptibilities there does not seem to be a strong trend in susceptibility across the country. However, Table 4 also provides potentially useful information to inform disease control efforts by identifying the land use classes (broadleaved and agricultural) that contribute most strongly to susceptibility and infectivity. Such information may be useful in guiding development of control policies and programmes.

In addition, risk maps can be helpful for visualising spatial predictions of which areas will become infected over defined scenarios and specified time horizons. Based on observation of the early stage epidemic, up to September 2015 (grey patches), Fig 8B shows model predictions of the probability of future infection. Such information can aid decisions about where to focus limited resources for disease control as the outbreak unfolds in real time. This prediction is over a specified time horizon under the assumption that rates of spread remain fixed. However, we also note that such use of this information means that actual spread will ideally be different from that predicted. This is due to the impact of disease controls as illustrated in Fig 7C–E which suggests a substantial benefit of disease control onwards from 2016.

By comparing spatial risk prediction to actual case reports over a limited time horizon we see the heterogeneous model identifies future infected patches slightly better than the homogeneous model (Table 5). However, the differences between the two models' predictions are not substantial. One reason for this observation is likely to be that the landscape

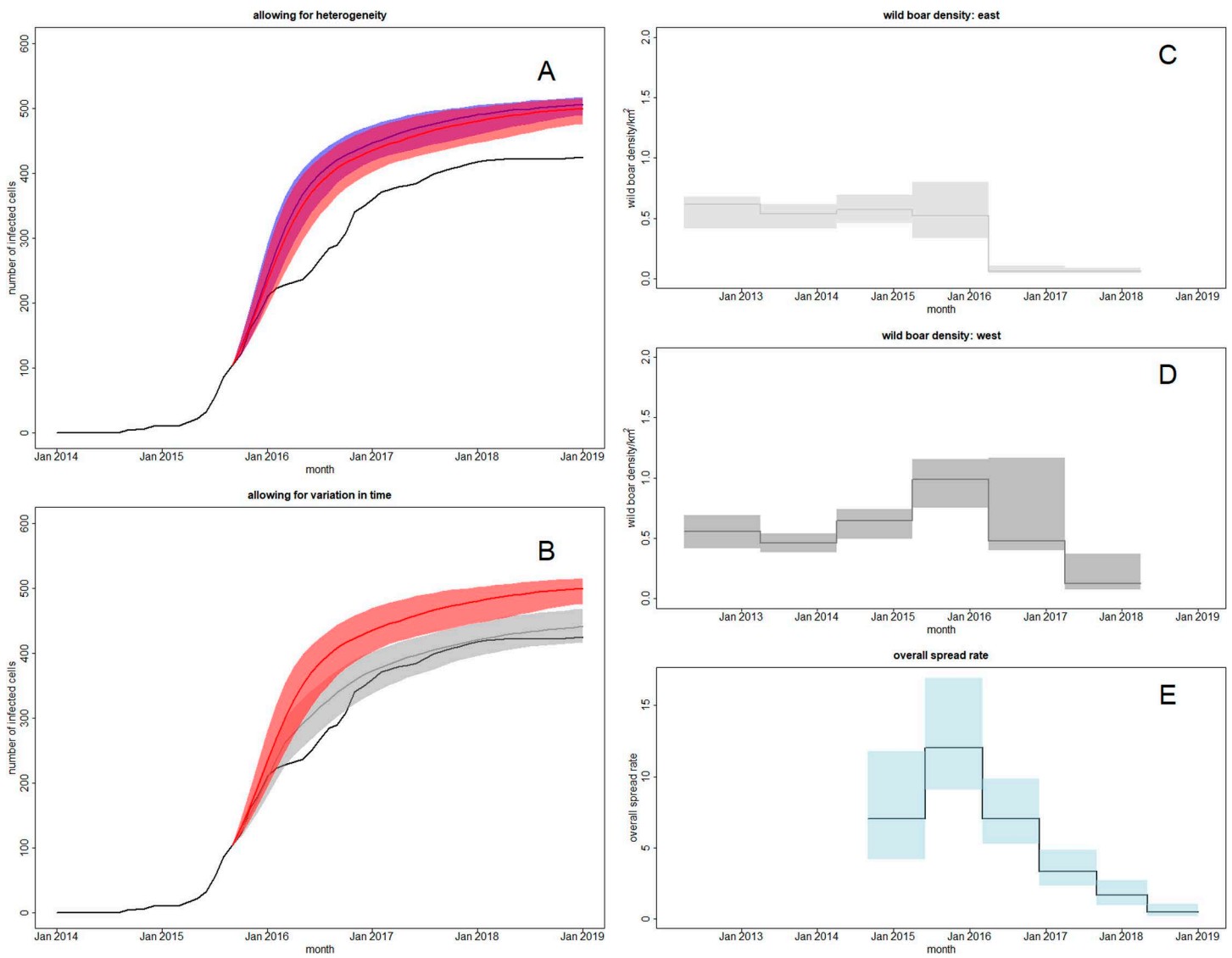

**Fig 7. Model prediction and validation.** *Left column*: Predicted numbers of infected 10km patches from models fitted to the early stages of the ASF outbreak in Estonian wild boar. Two constant-in-time models (heterogenous, homogeneous) were fitted to outbreak data up to September 2015. The black lines above show the true cumulative incidence curve at 10km patch resolution. A: the blue line and region show the posterior predictive median and 95% credible interval for the total number of infected patches under the homogeneous model. The red line and region show the corresponding quantities under the heterogenous model. B: the red line and region show the posterior predictive median and 95% credible interval for the total number of infected patches under the heterogeneous model. The grey line and region show the corresponding quantities under a varying-in-time heterogeneous model fitted to the full set of outbreak data. *Right column*: External validation via hunting bag data from [40]. C: estimated wild boar density in the east of Estonia. D: estimated wild boar density in the west of Estonia. E: estimated overall spread rate $\rho h(t)$ from the varying-in-time model fitted to the entire ASF outbreak in Estonia. The black line represents the posterior median, the blue area shows the 95% credible interval.

of Estonia is relatively homogeneous in the sense that there is little spatial variation in susceptibility at large scales (see Benefits of modelling spatial variation in susceptibility and infectivity in Results). Another explanation may be the narrowness of the estimated kernel, which is much narrower than the kernels used in Benefits of modelling spatial variation in susceptibility and infectivity in Results. Further simulations (S10 Appendix) similar to those in Benefits of modelling spatial

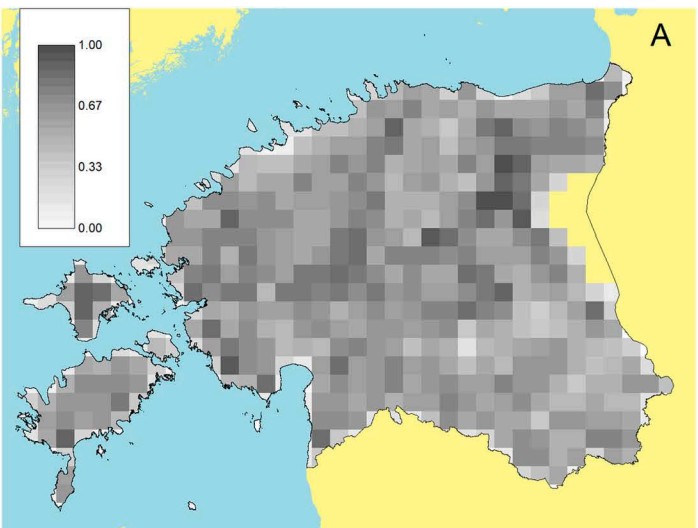
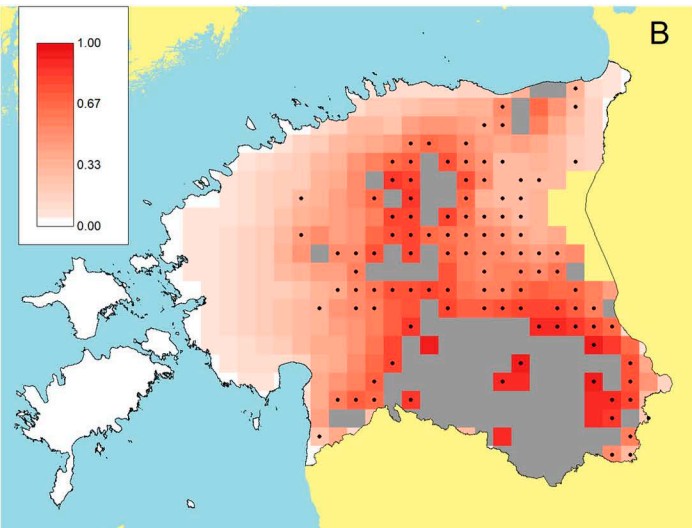

**Fig 8. Quantifying disease risk across Estonia.** (A) Susceptibility map for Estonia obtained from fitting the constant-in-time model to the Estonian ASF wild boar outbreak data. (B) Risk map illustrating predictions from the constant-in-time heterogeneous model fitted to the early stages of the ASF outbreak in wild boar in Estonia. Grey patches are recorded as infected at the end of the early stage of the outbreak. Red colours indicate probability of infection four months beyond this end point. The black dots show patches that did in fact become infected within this time horizon. Alternative summaries of the same predictions may be found in Table 5 and Fig 7A. Map base layer taken from https://gadm.org/maps.html.

**Table 5. Predictive performance of the fitted constant-in-time model.**

| | Discrimination metrics | | $N_{est}/N_{truth}$ | |
|---|---|---|---|---|
| **Model** | **AUC** | **TPR @ 5% FPR** | **2.5%ile** | **97.5%ile** |
| homog | 0.90 | 0.44 | 0.97 | 1.39 |
| heter | 0.90 | 0.47 | 0.92 | 1.34 |

Performance of the constant-in-time model fitted to data from the early stages of the ASF outbreak in wild boar in Estonia. For each fitted model (homogeneous/heterogeneous) the table shows AUC (area under the ROC curve) and the true positive rate at 5% false positive rate (TPR @ 5% FPR). $N_{est}/N_{truth}$ is the predicted number of infected patches divided by the true number of infected patches. All metrics were evaluated at a time of four months beyond the last observations used for fitting the models.

variation in infectivity and susceptibility in Results but with a narrow kernel give predictive performance similar to that in Table 5.

## Discussion

This paper develops the novel iPAR (inference for populations at risk) framework for inferring and modelling spread of disease in an unknown population at risk, given outbreak data in the form of only case reports with possibly less than precise spatial and temporal coordinates. The methodology introduced thus also contributes to tools suitable for scenarios where anonymisation of data are required. The method introduced exploits the intuitive idea that case reports simultaneously provide information on both disease spread and the population at risk, enabling modelling in relatively data poor scenarios not previously amenable to inference based on mechanistic transmission models. Where it is available the iPAR framework also allows for use of spatial covariate data that may be informative about the population at risk. iPAR uses a patch-based model with patch susceptibility and infectivity varying across the region over which disease is spreading. By

focussing on estimation of patch susceptibilities and infectivities we circumvent any requirement for detailed knowledge of the spatial distribution of the population at risk. In particular, we emphasise that the patch susceptibilities/infectivities are not population density estimates, but they should account for at least some of the variation in population density. If spatial population density estimates are available then they could be used as covariates in the model, possibly with a strong prior attached to the associated effect if we believe that the estimates are particularly informative.

As discussed in the Introduction, other methods have been developed to handle situations where there is uncertainty regarding the population at risk. Transmission tree models may be better suited to questions of "who infected who?" but are not suited to making predictions of future spread. Contact distribution models are better suited to making predictions, and it would be interesting to compare these models to the iPAR approach, but note that only the latter accounts for depletion of susceptibles. Models based on prediction of host (e.g., farm) locations can also be used to predict future spread, but in such cases modelled host locations are based solely on prior knowledge unlike in the iPAR approach where disease cases inform the inferred susceptibility and infectivity surfaces, which may correlate with host density but are directly informative of transmission processes.

We demonstrate, via a range of simulation studies, that the inference techniques employed here reliably estimate model parameters, with uncertainty reflecting available case data. Typically, parameters in the transmission kernel, as well as external transmission rates, are well estimated, susceptibility parameters are moderately well estimated, while as is to be expected infectivity parameters are the least well estimated, e.g., see [36]. It is challenging to precisely define the level of outbreak data availability needed to robustly estimate model parameters. Simple spatial spread models with no variation between infectious units can be fitted even when there are less than 10 reported disease cases [14]. Larger numbers of cases may be required for precise estimates when fitting more complex models. Analytic expressions for the precision of parameter estimates have been derived for non-spatial models that allow for variation in susceptibility and infectivity [43]. To successfully fit the iPAR model, the required number of cases will depend on the model structure and choice of spatial covariates. For the scenarios considered in this paper, we found that – as a very rough rule of thumb - between 10 and 100 cases are required to obtain robust parameter estimates. The duration of time over which the cases used to fit the model are collected is of lesser importance than the number of cases.

The iPAR approach is also able to estimate the 'overall' rate of infection $\rho$, and time variation in this quantity captured by $\rho h(t)$. In the case study this estimated time variation was shown to correlate with impacts of disease control efforts to reduce wild boar density in Estonia. Application of the iPAR approach is thus able to retrospectively characterise an outbreak and assess likely benefits of control actions taken. However, perhaps more critically iPAR can also be used to provide information useful for risk assessment during an outbreak. In the ASF case study presented here this includes: (i) identifying which land use classes and thus areas pose the highest intrinsic disease risks – potentially enabling targeted controls; (ii) quantifying the scale of spatial spread – information that could be used to design protocols for exclusions zones, ring vaccination or culling; and (iii) spatio-temporal quantification of future risk – identifying when and where new cases are most likely to arise thereby directing effort to where it is most needed in real time.

A key innovation of the iPAR framework is the ability to account for variation in susceptibility and infectivity across the landscape. Simulations highlighted scenarios where the heterogeneous model, with varying susceptibility/infectivity, significantly outperforms the corresponding homogeneous model. However, in our case study the heterogeneous model offered only a small improvement over the homogeneous model, but was nonetheless favoured by model selection. It is possible that use of other covariates not available to our analysis, might yield a heterogeneous model that offers a greater improvement, but the narrowness of the estimated kernel (short range transmission) makes this less likely. Intuitively, with a narrow kernel, there are fewer possible destinations for onward transmission of disease, so having a heterogeneous model that attempts to distinguish these possible destinations becomes less useful.

The most appropriate spatial resolution at which to model the spread of disease is dependent on multiple application-specific factors. If the spatial resolution is too low then the model may not be able to produce accurate

estimates of quantities of interest [44], but if the resolution is too high then there may be computational issues and a lack of suitable outbreak/covariate data. Other factors to consider are the intended application, model assumptions and the key quantities of interest. In the case study presented here, the *SI* model assumes persistence of infection, which is less likely to be reasonable at very high resolutions. We chose 10km resolution for the case study as analysis demonstrated good persistence at this scale. Furthermore, preliminary analysis of the case study data at 5km resolution gave similar results (S11 Appendix) suggesting considerable robustness of results to patch-scale.

To handle model parameters constrained to the simplex, e.g., the susceptibility parameter $\sigma$, this work made novel, within the context of transmission models, use of methods from compositional data analysis, such as the isometric logratio transform. As noted previously, there is typically slight overestimation of the 'overall' rate of infection $\rho$, which is proportional to the rate of transmission between two uniform patches (patches containing equal proportions of all land uses). Investigation of this phenomena suggests that it arises from the influence of the prior distribution on parameters for which the available data are not strongly informative (S7 Appendix). However, we note here that the properties of transmission models containing such constrained parameters do not seem to have been well explored in the epidemic modelling literature. Issues for further research include specification of prior distributions for parameters on the simplex, and interpretation of credible intervals for parameter components that are highly constrained and interdependent. In scenarios where knowledge of the population at risk is limited, compositional covariates such as land use proportions may be the only data that we have available, so methods that make best use of such data could have significant practical impact on disease control in data poor scenarios such as those addressed here.

Future methodological developments could extend the implementation of the iPAR framework to include more general patch-level disease dynamics beyond the susceptible-infected dynamics employed here. For example, a SEIR model could be implemented, but this would depend on either sufficiently informative data or strong prior assumptions. It would also be possible to allow for uncertainty in observations by treating them as imperfect diagnostic tests, but note that impact of false negative observations are somewhat mitigated in the iPAR framework, with both patch-scale and interval censoring reducing the probability that patches with significant infection remain unobserved. The Bayesian inference is currently based on a likelihood approach. The use of a more complex transmission model might increase the computational requirements, possibly necessitating a change to an approximate inferential approach, though this would depend on the specifics of the model and data.

It would also be desirable to implement iPAR for multiple host species, e.g., transmission of ASF in wild boar and domestic pigs, or transmission of *Xylella fastidiosa* in multiple susceptible plant species. Nonetheless there are numerous potential applications of the implementation of the iPAR framework introduced here. These include not only the study of other outbreaks of ASF in wild boar and/or domesticated pigs, but also other transmissible disease outbreaks where case only data are available. These might include Newcastle disease in poultry or highly pathogenic avian influenza in wild birds and/or backyard poultry.

In conclusion, we have developed a modelling framework that fills an important gap in epidemiological modelling. The iPAR approach allows modelling of spatial disease spread when the distribution of the disease host is uncertain, and the model can be parameterised using only disease case report data. In recent years, the amount of data available to inform models for spatial disease spread has rapidly increased. At the same time, data availability and data privacy continue to be an issue in many situations. Our approach can circumvent the requirement for host distribution data and so can open up many disease outbreak datasets for analysis.

## Supporting information

**S1 Appendix. Kernel normalisation.**
(DOCX)

**S2 Appendix. Prior distributions.**
(DOCX)

**S3 Appendix. Likelihood and sampling algorithm.**
(DOCX)

**S4 Appendix. DIC computation and values.**
(DOCX)

**S5 Appendix. The simplex and the isometric logratio transformation.**
(DOCX)

**S6 Appendix. Additional figures for Estimation of key epidemiological parameters in Results.**
(DOCX)

**S7 Appendix. Overestimation of the transmission rate $\rho$.**
(DOCX)

**S8 Appendix. Details of the setup for simulations in Benefits of modelling spatial variation in susceptibility and infectivity in Results.**
(DOCX)

**S9 Appendix. Persistence of infection analysis.**
(DOCX)

**S10 Appendix. Narrow transmission kernel simulations.**
(DOCX)

**S11 Appendix. Spread of ASF in Estonia at different spatial scales.**
(DOCX)

## Author contributions

**Conceptualization:** Stephen Catterall, Thibaud Porphyre, Glenn Marion.

**Data curation:** Thibaud Porphyre.

**Formal analysis:** Stephen Catterall, Glenn Marion.

**Funding acquisition:** Stephen Catterall, Thibaud Porphyre, Glenn Marion.

**Investigation:** Stephen Catterall, Thibaud Porphyre, Glenn Marion.

**Methodology:** Stephen Catterall, Thibaud Porphyre, Glenn Marion.

**Software:** Stephen Catterall.

**Validation:** Stephen Catterall.

**Visualization:** Stephen Catterall.

**Writing – original draft:** Stephen Catterall, Thibaud Porphyre, Glenn Marion.

**Writing – review & editing:** Stephen Catterall, Thibaud Porphyre, Glenn Marion.

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
