## [Decision Letter · Decision Letter 0]

PCOMPBIOL-D-24-01946

iPAR: A framework for modelling and inferring information about disease spread when the populations at risk are unknown

PLOS Computational Biology

Dear Dr. Catterall,

Thank you for submitting your manuscript to PLOS Computational Biology. After careful consideration, we feel that it has merit but does not fully meet PLOS Computational Biology's publication criteria as it currently stands. Therefore, we invite you to submit a revised version of the manuscript that addresses the points raised during the review process.

Please submit your revised manuscript within 60 days Mar 23 2025 11:59PM. If you will need more time than this to complete your revisions, please reply to this message or contact the journal office at ploscompbiol@plos.org. Please include the following items when submitting your revised manuscript:

We look forward to receiving your revised manuscript.

Kind regards,

Nicholas Geard

Academic Editor

PLOS Computational Biology

Denise Kühnert

Section Editor

PLOS Computational Biology

**Additional Editor Comments:**

The reviewers appreciated the importance of the problem you are addressing, and innovative nature of your proposed solution. They also provided detailed suggestions for improving the clarity and utility of your manuscript by including additional discussion of:

factors such as data availability, selection of covariates, computational cost, that may be relevant to use of the proposed methods by other researchers.the applicability of the proposed methods to populations and diseases with different characteristics (eg, human populations, recovery/immunity)how your approach compares to existing modelling methods.

While the provision of code was appreciated, clearer documentation would make your methods more readily usable by others.

**Journal Requirements:**

At this stage, the following Authors/Authors require contributions: Stephen Catterall, Thibaud Porphyre, and Glenn Marion. Please ensure that the full contributions of each author are acknowledged in the "Add/Edit/Remove Authors" section of our submission form.

5) We notice that your supplementary Figures, and Tables are included in the manuscript file. Please remove them and upload them with the file type 'Supporting Information'. Please ensure that each Supporting Information file has a legend listed in the manuscript after the references list.

Potential Copyright Issues:

- Figures Figures 6, 8, A10, A11, and A12. Please provide a direct link to the base layer of the map (i.e., the country or region border shape) and ensure this is also included in the figure legend; and provide a link to the terms of use / license information for the base layer image or shapefile. We cannot publish proprietary or copyrighted maps (e.g. Google Maps, Mapquest) and the terms of use for your map base layer must be compatible with our CC BY 4.0 license.

7) Please ensure that the funders and grant numbers match between the Financial Disclosure field and the Funding Information tab in your submission form. Note that the funders must be provided in the same order in both places as well.

**Reviewers' comments:**

Reviewer's Responses to Questions

**Comments to the Authors:**

Reviewer #1: My comments are uploaded as an attachment

Reviewer #2: The manuscript introduces the iPAR (inference for populations at risk) framework, a novel modeling and inference approach for spatial infectious disease dynamics when the population at risk is uncertain or poorly quantified. This is a non-trivial and important problem in epidemiology and population health, where precise host distribution data are frequently unavailable. The proposed approach is both conceptually interesting and methodologically innovative. It attempts to extract critical epidemiological information—such as susceptibility, infectivity, and rate of spread—from case-only data, supplemented by spatial covariates. The manuscript includes a theoretical and methodological development, as well as simulation-based evaluations and a real-world case study on African Swine Fever (ASF) in Estonian wild boar populations.

# Strengths:

The introduction of the iPAR framework addresses a critical gap in spatial disease modelling, especially when populations at risk are poorly quantified. This framework is highly relevant for wildlife outbreaks and under-documented livestock settings.

The step-by-step development of the model, including Bayesian inference, patch-based structure, and the use of covariates, is thoroughly detailed.

The authors show that the method can incorporate spatial covariates—land use, climate factors, or other relevant environmental proxies—to estimate susceptibility and infectivity surfaces. The framework is not restricted to a specific host-pathogen system and can be extended, at least in principle, to more complicated compartmental structures or multiple host species.

# Potential limitations and suggestions (major comments):

While the paper acknowledges the difficulty of recovering infectivity parameters compared to susceptibility, it may be beneficial to discuss conditions under which identifiability issues arise more explicitly. For instance, what levels of data availability (in terms of number of cases, spatial resolution, and outbreak duration) are needed for robust inference of both susceptibility and infectivity?

The authors note that the method can use a wide range of covariates. However, more guidance on how to select appropriate covariates, how to handle spatio-temporal variation in these covariates, and how to choose the patch size would strengthen the applicability. While patch-based models are common, best practices or decision criteria for resolution selection would be useful, especially in more heterogeneous landscapes or when data resolution differs between outbreak reports and covariates.

The current work focuses on an SI-type model with persistence. Many real-world diseases involve recovery, waning immunity, or multiple host types. A clearer roadmap on how to incorporate these complexities within the iPAR framework would strengthen its applicability. While mentioned as a future direction, more concrete guidance or at least a detailed conceptual outline would improve the paper’s practical utility.

In addition, clearly state the implications of using the SI model, especially for diseases like ASF that have latent phase, or other that would have a recovery phase.

Moreover, since inference is based on a likelihood-based approach, the use of a more complex model seems likely to severely hamper the use of the proposed framework. It is essential at the start of the article to clearly define the conditions of application of the framework.

The methods may be computationally intensive for large-scale problems with many patches and long observation periods. Although the paper mentions computational tractability as a motivation for patch aggregation, a discussion of computational scaling or potential performance optimizations (e.g., parallelization, approximation methods) would be valuable. No indication is given of the language used or the computing resources consumed (time, memory), which would give a better idea of the framework's performance and potential for improvement.

In "2.2.3 Uncertainty in model predictions" : "...initial conditions typically set to the infection status of the patches at the final time point of the

data used to fit the model ( = ). ...". In this case, it's assumed that the system is perfectly observed, but it's highly unlikely, and often impossible, to observe this kind of system exhaustively. Why not use as initial conditions the state of the system at t=T obtained in the simulations carried out? If the data do not contain an observation of all the patches, then it is very likely that the assumption made about the initial conditions tends to minimize the number of infected patches.

In "2.2.4 Testing the iPAR modelling approach" : This type of approach (the use of synthetic/simulated data) also has its limitations and drawbacks, which should be mentioned and discussed (choice of synthetic trajectory, etc.).

In "3.1 Estimation and prediction for an illustrative simulated outbreak" and "3.2 Estimation of key epidemiological parameters" : In these sections, no mention is made of how the simulated trajectory used as an observation was chosen, even though this can have a significant impact on the results. It should be described and discussed.

The approach infers “effective” susceptibility and infectivity surfaces that combine unknown population density and other factors. This layered interpretation may cause confusion for end-users. More explicit statements about the interpretative limits of these inferred surfaces, and how to use them alongside or in the absence of independent population density estimates—would help ensure that the results are not misinterpreted as direct host density maps.

While the paper’s accompanying code is appreciated, it appears incomplete. It lacks clear guidance on compilation and execution (notably due to the C code), and there are no provided examples or instructions for the necessary input files. More comprehensive documentation is mandatory.

# Minor comments:

- Define "external transmission" explicitly to ensure clarity for readers less familiar with disease modelling.

- Figures are informative, but their descriptions could be more concise. Ensure that captions focus on the key message of each figure (e.g., “parameter recovery demonstrates…”). For few figures, the addition of a legend would make reading easier.

- Part of the penultimate paragraph of the introduction (first half) seems to belong to the second part, which focuses on method.

- Tables : In the definition of scenarios, the "Susceptibility" variable always seems to be the same, and therefore not very useful to have it in the tables.

# Summary:

It's an interesting paper with valuable insights, but it is quite lengthy and dense, which can make it challenging for the reader to maintain a clear grasp of the material, and make it easy to occasionally lose track of the main thread, but well worth reading.

This manuscript makes a valuable contribution to spatial disease modelling, particularly for scenarios where host population data are limited or unavailable.

Addressing the suggested clarifications on model limitations, assumptions, guidance, and extended discussion on model complexity would further enhance the manuscript.

Reviewer #3: iPAR: A framework for modeling and inferring information about disease spread when

the populations at risk are unknown

comments:

The authors developed the iPAR framework which enables modeling and estimating of the spatial disease dynamics when the populations at risk are unknown. The authors presented rigorous simulation analysis with different spatiotemporal scenarios and real-world data analysis results. It is very interesting and relevant work since there are many situations we need to make strong assumptions about susceptible populations. This work can contribute to the infectious disease literature providing the modeling framework with a more accurate and flexible approach when we have the incomplete population distribution. However I have a few questions regarding the iPAR framework described in the manuscript.

1. I had the impression that the iPAR framework targets the infectious disease of animals, even though the manuscript did not specifically mention it. Can it be applied to human infectious disease transmission modeling? Is there any appropriate scenario of human infectious disease that the iPAR framework can be well suited? If it can be, I think it is necessary to show i) how iPAR model terms are changed or similarly used in human infectious disease transmission modeling, ii) which covariates can be applied to estimate surface infectivity and susceptibility, in case of human transmission, iii) how it can be used to model and predict infectious human disease transmission in simulation analysis. Currently, the model covariates ( e.g. Land use), simulation scenarios and real-world applications all focus on animal infectious disease modeling. If not, the manuscript needs to specifically mention that this paper mainly targets animal infectious disease transmission to help better understanding of audiences.

2. External Validation: This is a novel framework to model infectious disease transmission when the population at risk is unknown. As the authors wrote in section 1, there exist previous modeling methods based on transmission trees, contact-distribution models, and approaches based on hypothesized population distributions. What will be the benefits of using the iPAR framework compared to existing approaches? How will model performance be different ( e.g. DIC, TPR and FPR) among different approaches? In which simulation scenario does the iPAR perform better than other existing approaches? The authors can show it with different simulation analyses.

3. The iPAR framework uses case reports of the region, but it only estimates and predicts whether the patch itself is infected or not. Is the small/ large number of cases in each patch associated with the performance of the iPAR framework? Or is it only a binary decision that is important here (patch is infected at time t, or not)? It would be a relevant question whether the performance of the iPAR framework will be the same for the regions (multiple patches combined) with only a small number of cases and with a large number of cases, similar to the small area estimation problem in aggregated spatial entities. Will the performance metrics or the width of 95% credible intervals of estimators be the same for these areas with small and large cases? Simulation or real-world data analysis can show this difference.

Reviewer #4: While the subject of the study is of great interest and would benefit readers, more work is required on the presentation of the manuscript. Overall, the manuscript provides a mathematically robust and flexible framework for dealing with missing population data in spatial epidemiology.

However, my initial impression is that the manuscript is too lengthy (25 pages of the manuscript and an additional 25 pages containing 10 appendices), which may divert readers' attention. The manuscript would greatly benefit from an effort by the authors to shorten the text, summarize the appendices within the main text, and integrate the explanation of the model’s application with case study and parameter estimation, rather than presenting them in separate sections.

The text is logically coherent and maintains a consistent framework throughout, but the exposition could be streamlined for clarity. Most sections are dense and may be challenging for readers without a strong mathematical background. Simplification would benefit the overall readability. For example, the likelihood expression and Bayesian updates, while technically accurate, are dense and could overwhelm readers unfamiliar with Bayesian modeling or epidemic processes.

The paper would benefit from merging the explanation of the model with its application, both in the case study and the estimation of key parameters, rather than separating them into distinct sections. This would improve the flow and make it easier to understand the model’s real-world implications.

The manuscript demonstrates a flexible approach to addressing uncertainties in the spatial distribution of populations at risk. By relying on covariates, the model accommodates varying levels of data availability, making it adaptable to scenarios where detailed population data is missing. However, the effectiveness of this flexibility depends on the quality and relevance of the covariates used. For this study, only land use was employed as a covariate, but further discussion on the selection and justification of covariates could enhance the model's applicability.

The methodology is mathematically specified with clear equations, but reproducibility in practice could depend on the following aspects:

Details on the implementation and parameter estimation, including the software used.

Explanation of how covariates are pre-processed, standardized, or selected (i.e., defining what constitutes a "suitable covariate").

Simplification or rephrasing of the explanation and interpretation of susceptibility, infectivity, covariate-based functions, and parameters to make them more approachable.

While the incorporation of absolute and relative covariates is mathematically valid and aligns with common practices in modeling spatial heterogeneity, clarification is needed. For instance, what defines temperature as absolute and land cover as relative? If only land cover is used in the application, the entire explanation of absolute and relative covariates could be removed, keeping only the relevant covariate used in the model.

Clear explanations of what each parameter represents, the rationale behind the values or algorithms used to estimate them, and the references used to assign these values.

The choice of non-informative priors requires further justification. Informative priors could be incorporated based on relevant epidemiological, spatial, and temporal factors. For example, proximity to known sources (infection time may depend on the distance from other infected patches), wild boar or farm density (higher densities may lead to earlier infections), patch-level biosecurity (better biosecurity could delay infection), seasonality (ASF transmission may exhibit seasonal trends), outbreak start time (infection times should logically follow the initial outbreak in nearby patches), and ASF's tendency to spread via contact (e.g., wild boar migration, human-mediated transmission). Additionally, models for spatial spread (e.g., diffusion or gravity models) could inform the selection of priors.

Other minor comments:

Acronyms should be spelled out the first time they are mentioned in the manuscript (e.g., MCMC). The term "ESM" can be removed and simply refer to the appendix number.

Incorporating line numbers would be a useful practice for facilitating specific revision comments.

Ensure that references are cited correctly in the text. For example, on page 6, it should read: O'Neill and Roberts (1999), Jewell et al. (2006), and Stockdale et al. (2017). Revise all reference citations accordingly.

When citing other authors, summarize their arguments clearly to provide context to the reader.

**Have the authors made all data and (if applicable) computational code underlying the findings in their manuscript fully available?**

Reviewer #1: Yes

Reviewer #2: Yes

Reviewer #3: Yes

Reviewer #4: **No: ** The data was downloaded from public websites. No code was provided, nor details of software used.

PLOS authors have the option to publish the peer review history of their article (what does this mean? ). If published, this will include your full peer review and any attached files.

**Do you want your identity to be public for this peer review?** For information about this choice, including consent withdrawal, please see our Privacy Policy .

Reviewer #1: No

Reviewer #2: No

Reviewer #3: No

Reviewer #4: No

**Figure resubmission:**

**Reproducibility:**



---

## [Decision Letter · Decision Letter 1]

Dear Dr Catterall,

We are pleased to inform you that your manuscript 'iPAR: A framework for modelling and inferring information about disease spread when the populations at risk are unknown' has been provisionally accepted for publication in PLOS Computational Biology.

Reviewer 2 has made two further suggestions. I leave to your discretion whether you wish to incorporate these in your manuscript.

Best regards,

Nicholas Geard

Academic Editor

PLOS Computational Biology

Denise Kühnert

Section Editor

PLOS Computational Biology

Reviewer's Responses to Questions

**Comments to the Authors:**

Reviewer #1: The modifications made by the authors in response to all reviewers' comments look satisfying to me, and importantly, the guide produced to re-use the code is very complete.

Reviewer #2: Major comments have been addressed; however, two points remain to be addressed:

- Manuscript length and organization: the main text remains dense; consider merging parts of results and case study to improve narrative flow.

- Perfect-observation assumption (p 6, l 22–24) may be unrealistic. Even if it is sometimes assumed, it is a characteristic that is very often taken into account, due to a potentially significant bias.

Reviewer #3: The authors provided the reasonable explanation for my previous comments.

**Have the authors made all data and (if applicable) computational code underlying the findings in their manuscript fully available?**

Reviewer #1: Yes

Reviewer #2: Yes

Reviewer #3: None

PLOS authors have the option to publish the peer review history of their article (what does this mean? ). If published, this will include your full peer review and any attached files.

**Do you want your identity to be public for this peer review?** For information about this choice, including consent withdrawal, please see our Privacy Policy .

Reviewer #1: No

Reviewer #2: No

Reviewer #3: No

---

## [Editor Report · Acceptance letter]

PCOMPBIOL-D-24-01946R1

iPAR: A framework for modelling and inferring information about disease spread when the populations at risk are unknown

Dear Dr Catterall,

I am pleased to inform you that your manuscript has been formally accepted for publication in PLOS Computational Biology. Your manuscript is now with our production department and you will be notified of the publication date in due course.

With kind regards,

Zsofia Freund
